# CaReBench: A Fine-Grained Benchmark for Video Captioning and Retrieval

**Yifan Xu[1], Xinhao Li[1], Yichun Yang[1], Desen Meng[1], Rui Huang[1], Limin Wang[1,2]***

[1]State Key Laboratory for Novel Software Technology, Nanjing Univerisity
[2]Shanghai AI Laboratory
`yifanxu@smail.nju.edu.cn, lmwang@nju.edu.cn`

## Abstract

Video understanding, including video captioning and retrieval, is still a great challenge for video-language models (VLMs). The existing video retrieval and caption benchmarks only include short descriptions, limits their ability of detailed video understanding evaluation. To address this problem, we present CaReBench, a testing benchmark for fine-grained video **Ca**ptioning and **Re**trieval with 1,000 high-quality pairs of videos and human-annotated detailed captions. Uniquely, it provides manually separated spatial annotations and temporal annotations for each video. Based on this design, we introduce two evaluation metrics, ReBias and CapST, specifically tailored for video retrieval and video captioning tasks, respectively. These metrics enable a comprehensive investigation into the spatial and temporal biases inherent in VLMs. In addition, to handle both video retrieval and video captioning tasks in a unified framework, we develop a simple baseline based on a Multimodal Language Model (MLLM). By implementing a two-stage Supervised Fine-Tuning (SFT), we fully unlock the potential of MLLM, enabling it not only to generate detailed video descriptions but also to extract video features. Surprisingly, experimental results demonstrate that, compared to the CLIP-based models designed for retrieval and the popular MLLMs skilled in video captioning, our baseline shows competitive performance in both fine-grained video retrieval and video detailed captioning. Project page: `https://carebench.github.io`

## 1 Introduction

Video captioning (Wang et al., 2022; Xu et al., 2023; Wang et al., 2024a; Chai et al., 2024) and video retrieval (Radford et al., 2021; Luo et al., 2022; Ma et al., 2022; Zhou et al., 2024; Wang et al., 2024d; Zhu et al., 2024; Zhang et al., 2024a) are two main tasks in video-language understanding. Captioning requires perception and description of the main objects, events and actions in the video, while retrieval aims at finding the most relevant video/text based on the text/video query. These two tasks intuitively reflect the alignment and comprehension ability of Video-Language Models (VLMs), serving as critical evaluations of VLM capabilities.

However, existing retrieval and captioning benchmarks still struggle to evaluate fine-grained understanding of VLM. Traditional benchmarks (Xu et al., 2016; Chen & Dolan, 2011; Hendricks et al., 2017) have short and rough annotations, assessing general and coarse-grained video understanding of VLMs due to brief descriptions. Recent works (Zhang et al., 2024a; Yang et al., 2024; Chai et al., 2024) use powerful VLMs like GPT-4o (OpenAI, 2023) for auto-annotation, which inevitably introduces hallucinations and potential biases. DREAM-1K (Wang et al., 2024a) has more accurate human annotations, yet it lacks hierarchical captions and truly comprehensive focus on both objects and events.

In addition, designing effective metrics for video captioning also poses a challenge. Traditional n-gram metrics (Vedantam et al., 2015) are difficult to evaluate fine-grained captions (Wang et al., 2024a; Chai et al., 2024), while LLM-based evaluations (e.g. AutoDQ (Wang et al., 2024a)), lack comprehensive consideration of both static objects and dynamic actions.

---

*Corresponding author: Limin Wang.

Table 1: **Statistics of retrieval and captioning benchmarks.** Traditional benchmarks, namely MSR-VTT (Xu et al., 2016), MSVD (Chen & Dolan, 2011), DiDeMo (Hendricks et al., 2017) and ActivityNet (Heilbron et al., 2015) have very short captions. Detailed captioning benchmarks (Wang et al., 2024a; Chai et al., 2024) have longer and detailed captions, but they are either annotated by GPT or fail to focus on both static objects and dynamic actions.

| Benchmark | # Sample | Avg. Len. | Avg. Words | Annotator | Hierarchical Anno. | Static Object | Dynamic Action |
|---|---|---|---|---|---|---|---|
| MSR-VTT (Xu et al., 2016) | 1,000 | 15.01s | 9.41 | Human | ✗ | ✗ | ✗ |
| DiDeMo (Hendricks et al., 2017) | 1,037 | 53.94s | 29.11 | Human | ✗ | ✗ | ✗ |
| MSVD (Chen & Dolan, 2011) | 670 | 10.04s | 7.01 | Human | ✗ | ✗ | ✗ |
| ActivityNet (Heilbron et al., 2015) | 5,044 | 36.00s | 13.48 | Human | ✗ | ✗ | ✗ |
| DREAM-1K (Wang et al., 2024a) | 1,000 | 8.9s | 59.3 | Human | ✗ | ✗ | ✓ |
| VDC (Chai et al., 2024) | 1,000 | 28.18s | 500.91 | GPT | ✓ | ✓ | ✗ |
| **CAREBENCH** | 1,000 | 14.35s | 227.95 | Human | ✓ | ✓ | ✓ |

To address these issues, we present CAREBENCH, a fine-grained **Bench**mark for video **Ca**ptioning and **Re**trieval. It contains 1,000 videos with human-annotated detailed captions. Unlike images, video understanding tasks require models to understand both static scenes and dynamic actions. So we apply a hierarchical annotation scheme with each annotation covering four aspects: an overall summary, static object descriptions, dynamic action descriptions, and misc descriptions (e.g., filming style, camera movement, etc.). Such a design ensures each caption has sufficient details, challenging models to capture fine-grained information. Furthermore, to evaluate models spatiotemporally, each caption is manually separated into spatial and temporal parts. Based on this, we construct ReBias and CapST, two novel metrics for video retrieval and captioning, respectively. Due to our benchmark and metrics design, this work brings the community some new insights about spatiotemporal biases of state-of-the-art VLMs that other benchmarks may fail to reveal.

During the evaluation on both video retrieval and captioning tasks, we realize that previous works often treat retrieval and captioning as separate tasks, leading to the development of specialized models for each. Specifically, CLIP-based dual-encoder models have been well advanced for video retrieval, while Multimodal Large Language Models (MLLMs) have been tailored for video captioning. However, we discover that the two tasks can be unified and formulated as a mapping from the pixel space to a high-dimensional space: $\phi : \mathbb{R}^{T \times H \times W \times C} \to \mathbb{R}^{D}$ (either vocabulary space $\mathbb{R}^{D_v}$ or embedding space $\mathbb{R}^{D_e}$). This finding renders it feasible to address the gap between video retrieval and captioning.

Taking advantage of the unified architecture of MLLMs, we develop CARE, a simple and unified baseline for both detailed video captioning and fine-grained video retrieval. Specifically, our method involves a two-stage supervised fine-tuning (SFT). This makes it possible to generate video captions and discriminate video contents using only one model. The first stage aligns the model output to a fine-grained text space, by training the model using mixed LLaVA-Video-178k (Zhang et al., 2024c) and Tarsier (Wang et al., 2024a) recaptioned data. In the second stage, a text-only contrastive learning approach (Jiang et al., 2024b) is adopted to enable the MLLM to effectively perform cross-modal representations. As shown in Figure 1, our experiments indicate that, compared to CLIP-based retrieval models and MLLM captioning models, CARE achieves superior performance on CAREBENCH.

In summary, we make the following contributions:

**(1)** We introduce a novel fine-grained benchmark named CAREBENCH. It is designed for video retrieval and captioning, comprising 1,000 videos with high-quality human-annotated descriptions that provide sufficient video details. Each video features hierarchical descriptions ensuring comprehensive coverage, and manually split spatial/temporal captions. Based on this, we construct ReBias and CapST, two novel metrics designed for the video retrieval and captioning tasks, respectively. Such designs reveal new insights about spatiotemporal biases of VLMs that other benchmarks may ignore.

**(2)** We present CARE, a simple baseline for fine-grained video retrieval and captioning. By applying two-stage Supervised Fine-Tuning (SFT), we enable CARE to not only generate detailed video descriptions but also to extract video features. Our experiments show that, compared to the CLIP-based models designed for retrieval and the popular MLLMs skilled in video captioning, our baseline has competitive performance in both fine-grained video retrieval and detailed video captioning.

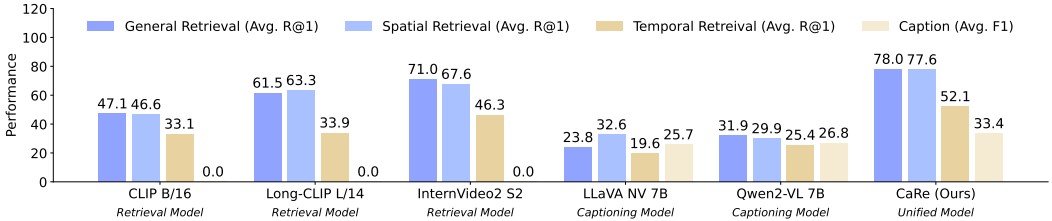

Figure 1: **CAREBENCH performance of popular models and CARE.** The results on MLLMs are reported on their public version without contrastive training. The CLIP-based models have achieved excellent performance in video retrieval tasks, but they lack the ability to describe videos. On the other hand, MLLMs can describe videos in detail, but their retrieval performance is very poor. In contrast, CARE not only shows outstanding retrieval performance but also has a strong capability to describe videos. Features are extracted from MLLMs using EOL prompt (Jiang et al., 2024b).

## 2 RELATED WORK

**Video Caption.** Video captioning aims to describe videos using natural language. Traditional captioning benchmarks, such as ActivityNet (Heilbron et al., 2015), MSVD (Chen & Dolan, 2011), and MSR-VTT (Xu et al., 2016), typically use a single sentence to describe a video, which is insufficient to convey the full visual contents. As a result, they can no longer effectively stress-test modern MLLMs, as these models can output semantically richer descriptions than reference captions. To address these issues, new benchmarks have been proposed. DREAM-1K (Wang et al., 2024a) annotates five categories of videos with rich action content and introduces a novel automatic evaluation method called AutoDQ, which assesses the accuracy and recall of actions and events in captions. Similarly, VDC (Chai et al., 2024) employs hierarchical prompting with GPT-4o for structured and detailed captions, followed by manual correction. However, it lacks explicit focus on human actions and motion. In this paper, we explore a new fine-grained video captioning benchmark focusing not only on objects but also actions to comprehensively evaluate VLMs.

**Video Retrieval.** Video retrieval aims to find the most relevant video/text based on the text/video query. Traditional methods (Wang et al., 2024d; Ma et al., 2022; Luo et al., 2022; Zhang et al., 2024a; Li et al., 2023; Girdhar et al., 2023) focus on using dual encoders based on CLIP (Radford et al., 2021) to extract features. But most of them are limited by the 77-token context length inherited from CLIP, hindering long-caption understanding (Zhou et al., 2024). While long-text and fine-grained video retrieval becomes important. Long-CLIP (Zhang et al., 2024a) addresses this problem by extending context to 248 tokens for long-text retrieval. But the benchmark used by it are annotated by LLMs, which may contain coarse-grained, uncertain and wrong descriptions. In this paper, we further explore the model training and the benchmark design for fine-grained video retrieval.

**Multimodal Large Language Model.** Due to great advancements in LLMs (Devlin et al., 2019; Brown et al., 2020; Wei et al., 2022; Chowdhery et al., 2023), their multimodal counterparts (MLLMs) (Li et al., 2025; Chen et al., 2023; Yao et al., 2024; Zhang et al., 2024b; Wang et al., 2024b) are receiving significant attention, particularly for their capability to perform various visual tasks using straightforward instructions. Recent works like VideoChat (Li et al., 2025), are trained on large-scale datasets such as (Wang et al., 2024c) and demonstrate outstanding performance on multimodal benchmarks (Fu et al., 2024; Li et al., 2024). But these models are restricted to generating responses based solely on user instructions and lack the capability to represent videos, images, and text. In this paper, we construct a unified baseline for both video retrieval and video captioning.

**Multimodal Embedding.** CLIP (Radford et al., 2021) learns image and text representations by aligning them with contrastive learning. However, Mind the Gap (Liang et al., 2022) notes that different data modalities are embedded with gaps in their shared representation space. To address this issue, recent works like VISTA (Zhou et al., 2024) and E5-V (Jiang et al., 2024b) explore unified representation. They find that MLLMs provide a unified multimodal framework to unify cross-modal representations without gaps. We regard it as a promising method and will further explore unified MLLM representation on video retrieval.

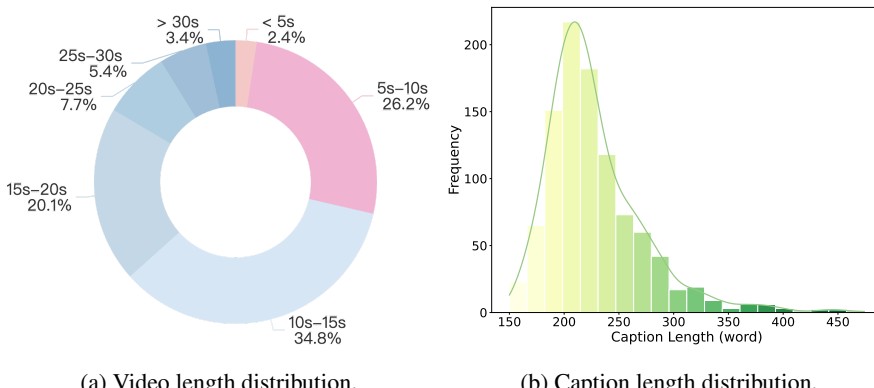

(a) Video length distribution.  (b) Caption length distribution.

Figure 2: **Statistics of CAReBench.** In our benchmark, most videos range from 5-20 seconds and most captions fall between 150 and 300 words in length.

## 3 CAReBench: a fine-grained benchmark

### 3.1 Video collection

We manually select 1,000 videos from FineAction (Liu et al., 2022) with 10-20 videos in each subcategory. FineAction is a video dataset for temporal action localization with 106 subcategories and 4 major categories: *personal care*, *socializing & relaxing*, *sports & exercise*, and *household activities*. Videos in each subcategory share similar scenes and actions, which poses a challenge to the models' ability to understand and discriminate similar videos.

### 3.2 Two-stage annotation pipeline

The annotation pipeline consists of two stages. In stage one, annotators describe videos in detail, covering four key aspects of each video. Subsequently, they are guided to separate the annotations into temporal and spatial descriptions. To ensure high quality and minimize bias, each video is independently captioned by two annotators and subsequently refined and merged by our experts. Refer to Figure 6 and Appendix A for annotation pipelines, data examples and the case study.

#### 3.2.1 Stage-I: detailed annotation

In Stage-I, annotators provide detailed video descriptions limited to 150-300 words. Each description can be divided into four parts: a general overview, an action description, a object description, and a misc. description, as outlined below:

**General Overview** provides a one-sentence summary of the entire video. For example, *this video shows a person slicing a watermelon.*

**Object Description** focuses on static objects with attributes like position, color, shape, and other visual details. It contains primary and secondary objects, background, their relative positions, interactions, and even watermarks.

**Action Description** captures the actions occurring in the video, detailing the event sequences (e.g., *first..., then...*) and providing specific details of each action (e.g., *rotating the watermelon clockwise*). It also includes the style of the actions (e.g., *cutting fruit quickly*, *climbing the tree clumsily*).

**Misc. Description** is about 2-4 sentences in length. It covers different aspects, such as the viewpoint (e.g., *a third-person perspective*) and the overall type of the video (e.g., *delightful and relaxing*).

#### 3.2.2 Stage-II: spatio-temporal separation

Stage-II refines the initial annotations by separating spatial and temporal elements. It removes action texts from object descriptions to create pure spatial captions, and eliminates static references

from action descriptions to form pure temporal captions. This design ensures precise evaluation of VLMs' spatiotemporal modeling capabilities by preventing interference between dynamic and static elements.

**Spatial Description** provides a comprehensive view, beginning with a general overview and then detailing main objects, secondary objects, and the backgrounds. It ensures that spatial descriptions can differentiate between similar videos within the same subcategory.

**Temporal Description** begins with a general overview, then focuses on actions and their order. Spatial-specific details are excluded. It ensures temporal descriptions uniquely identify each video within its subcategory.

## 3.3 COMPARISON ON STATISTICS

The captions in CAREBENCH are human-annotated, providing detailed and comprehensive descriptions of the videos. Consequently, its statistics differ significantly from those of traditional benchmarks. As shown in Table 1, our benchmark is similar in size to MSR-VTT (Xu et al., 2016), DiDeMo (Hendricks et al., 2017), but the average number of words per caption is 24.2× higher than that of MSR-VTT (Xu et al., 2016), 7.82× higher than DiDeMo (Hendricks et al., 2017), and 32.5× higher than MSVD (Chen & Dolan, 2011). The chart in Figure 2a shows the video length distribution of CAREBENCH. Since excessively long video durations significantly increase the difficulty for annotators to provide detailed descriptions, our benchmark focuses on videos ranging from 5 to 20 seconds in length, with over 80% of the videos falling within this range. Only 5.8% are shorter than 5s or extends beyond 30s. Figure 2b demonstrates how the caption length distributes. Most captions in CAREBENCH contain between 175 and 275 words.

## 3.4 METRICS DESIGN

CAREBENCH contains manually annotated highspatial-temporal captions. This design enables us to identify biases in the model's understanding of static objects and dynamic actions by analyzing the imbalance in spatio-temporal performance across video retrieval and captioning tasks. To quantify the spatio-temporal perfomance and bias, we introduce two novel metrics for video retrieval and video captioning, respectively: ReBias and CapST. These two metrics comprehensively reveal VLMs' performance and inherent biases by separately evaluating spatial tasks and temporal tasks.

### 3.4.1 REBIAS

Evaluating spatial and temporal captions separately reveals the model's performance across both dimensions. We introduce ReBias, a metric that measures spatiotemporal **Re**trieval **Bias**. It measures a model's bias towards its focus on static objects versus dynamic actions by showing how far the temporal-to-spatial recall ratio deviates from 1 (lower is better). It can be formulated as follows:

$$B = \left| 1 - \frac{\bar{R}_{temporal}}{\bar{R}_{spatial}} \right|, \tag{1}$$

where $\bar{R}_{temporal}$ and $\bar{R}_{spatial}$ denotes the average recall on temporal/spatial retrieval, respectively.

### 3.4.2 CAPST

Existing video captioning metrics face limitations: traditional n-gram methods, like CIDEr (Vedantam et al., 2015), struggle with long captions (Chai et al., 2024; Wang et al., 2024a), while LLM-based metrics (Chai et al., 2024; Wang et al., 2024a) lack comprehensiveness in evaluating both objects and actions. For example, VDCScore (Chai et al., 2024) evaluates predictions by querying both ground truth and prediction details to compute *recall*, but it ignores *precision* which is critical for assessing hallucinations; AutoDQ (Wang et al., 2024a) only focuses on evaluating actions/events and neglects objects. To overcome these issues, we propose CapST, a video **Cap**tioning metric jointly evaluating **S**patial objects and **T**emporal events. Similar to Wang et al. (2024a), a powerful LLM extracts events from temporal captions and objects from spatial captions and computes the Natural Language Inference (NLI) relationship between the ground truth $D_{gt}$ and the predictions $D_{pred}$. Specifically, we compute the recall and precision score of a sample according to Equation (3) and

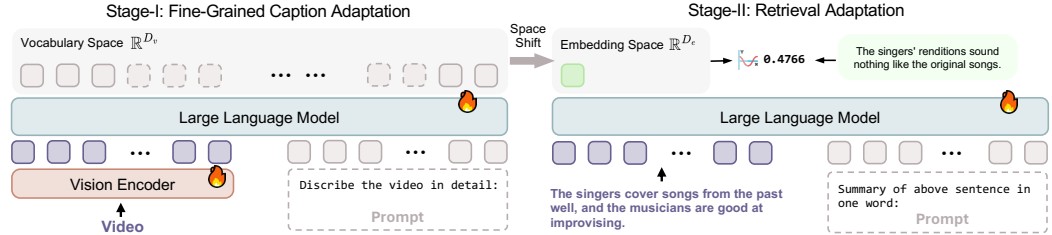

Figure 3: **Training recipe of CARE.** Stage-I aligns CARE outputs to a fine-grained text space for detailed video descriptions, while Stage-II contrastively trains CARE to extract features, shifting the output space from vocabulary ($\mathbb{R}^{D_v}$) to embedding space ($\mathbb{R}^{D_e}$).

report average recall and precision of all samples in a benchmark. More details about quantitative and human-aligned validation on different metrics can be seen in Appendix C.

$$R = \frac{N(D_{gt} \xrightarrow{\text{entail}} E_{pred})}{N(E_{pred})}, \tag{2}$$

$$P = \frac{N(D_{pred} \xrightarrow{\text{entail}} E_{gt})}{N(E_{gt})}, \tag{3}$$

where $E_{pred}$ and $E_{gt}$ denote elements (either objects or events) extracted from predictions and ground truth captions, respectively. $N(E_{pred})$ and $N(E_{gt})$ is the number of elements extracted from $D_{pred}$ and $D_{gt}$, respectively. $N(D_{gt} \xrightarrow{\text{entail}} E_{pred})$ refers to the number of $E_{pred}$ entailed by $D_{gt}$, and $N(D_{pred} \xrightarrow{\text{entail}} E_{gt})$ means the number of $E_{gt}$ entailed by $D_{pred}$.

Specially, when multiple attributes are combined in a single description (e.g., "*an elderly man wearing glasses and a blue suit*"), NLI evaluation tends to penalize partially matching predictions, even when those predictions correctly identify some valid characteristics. To address this issue, we instruct the LLM to split attributes during extraction. For instance, the aforementioned description would be divided into "*an elderly man wearing glasses*" and "*an elderly man wearing a blue suit.*" This design allows a more precise evaluation of the model's performance to describe objects with multiple attributes.

## 4 CARE: A UNIFIED VIDEO MODEL

Previous works treat video retrieval and captioning as separate tasks, fostering specialized models like CLIP-based dual-encoders for retrieval and MLLMs for captioning. However, we find that these tasks can be unified into a single framework, formulated as a mapping from the pixel space to a high-dimensional space: $\phi : \mathbb{R}^{T \times H \times W \times C} \to \mathbb{R}^D$ (either vocabulary space $\mathbb{R}^{D_v}$ or embedding space $\mathbb{R}^{D_e}$). To bridge this gap, we introduce CARE, a unified baseline built on Qwen2-VL (Wang et al., 2024b), trained via a two-stage progressive SFT to achieve both robust video captioning and strong video representation. The training pipeline is shown in Figure 3.

### 4.1 STAGE-I: FINE-GRAINED CAPTION ADAPTATION

MLLMs excel in general video understanding but often miss key video details. To align the model with fine-grained video understanding and provide a robust backbone for Stage-II, we train CARE with high-quality video-caption pairs. Specifically, we set finetuning prompt to "`Describe the video in detail.`" and train our model using video-text pairs from Tarsier Recap (Wang et al., 2024a), emphasizing action-rich descriptions, and LLaVA-Video-178k (Zhang et al., 2024c), focusing on short videos with details. With fine-grained caption adaptation, the model output is aligned with fine-grained text space and can better focus on detailed actions and objects when describing videos.

Table 2: **Video caption performance of popular models on CAREBENCH (Events).** We report F1/Recall/Precision for each category. # Params denotes the number of LLM parameters.

| Model | # Params | CAREBENCH Caption (Events) | | | | |
|---|---|---|---|---|---|---|
| | | Personal Care | Social & Relax | Sports & Excercise | Household | Overall |
| GPT-4o mini | - | 32.9/24.9/48.4 | 34.7/26.2/51.1 | 44.3/38.0/53.0 | 34.2/26.9/46.8 | 36.8/29.1/50.2 |
| LLaVA NeXT Video (Zhang et al., 2024b) | 7B | 27.5/20.1/43.7 | 25.0/17.4/44.1 | 29.4/21.1/48.4 | 24.3/16.2/48.1 | 26.6/18.7/45.9 |
| InternVL2 (Chen et al., 2023) | 7B | 22.2/18.4/28.0 | 23.0/17.9/32.3 | 27.9/23.4/34.5 | 18.4/14.7/24.8 | 23.3/18.8/30.7 |
| InternVL2.5 (Chen et al., 2024) | 7B | 22.0/15.1/41.1 | 24.0/16.8/41.6 | 34.0/26.1/48.8 | 22.3/15.3/40.6 | 26.0/18.6/43.2 |
| InternVL2.5 (Chen et al., 2024) | 72B | 24.6/16.7/46.7 | 25.9/18.3/44.4 | 36.0/27.8/51.0 | 24.9/17.5/43.2 | 28.2/20.3/46.4 |
| MiniCPM-V 2.6 (Yao et al., 2024) | 7B | 30.2/21.3/52.0 | 26.9/18.6/48.8 | 38.1/29.7/53.1 | 28.5/20.0/49.5 | 31.1/22.3/51.2 |
| Tarsier (Wang et al., 2024a) | 7B | 25.4/16.5/55.0 | 26.5/18.0/50.4 | 32.0/22.8/53.3 | 22.8/15.3/44.7 | 27.1/18.4/51.1 |
| Qwen2-VL (Wang et al., 2024b) | 7B | 28.4/23.9/34.9 | 27.5/20.8/40.3 | 33.0/26.6/43.6 | 25.7/20.2/35.1 | 28.8/22.9/39.0 |
| Qwen2-VL (Wang et al., 2024b) | 72B | 29.6/22.1/45.0 | 28.1/20.6/44.2 | 37.3/28.5/53.9 | 26.4/18.6/45.4 | 30.5/22.6/47.1 |
| Qwen2.5-VL (Bai et al., 2025) | 7B | 30.0/21.2/51.0 | 29.7/21.3/48.9 | 36.1/28.0/50.8 | 27.2/19.4/45.6 | 31.1/22.7/49.2 |
| CARE$_{\text{stage-I}}$ | 7B | 33.9/25.4/50.8 | 32.4/24.0/49.8 | 42.8/33.7/58.5 | 31.5/24.4/44.7 | 35.3/26.9/51.3 |
| CARE | 7B | 34.4/25.6/52.6 | 32.2/24.0/48.8 | 42.3/33.3/58.1 | 30.9/23.4/45.3 | 35.1/26.6/51.4 |

Table 3: **Video caption performance of popular models on CAREBENCH (Objects).** We report F1/Recall/Precision for each category. # Params denotes the number of LLM parameters.

| Model | # Params | CAREBENCH Caption (Objects) | | | | |
|---|---|---|---|---|---|---|
| | | Personal Care | Social & Relax | Sports & Excercise | Household | Overall |
| GPT-4o mini | - | 29.2/21.2/47.2 | 34.2/26.5/48.0 | 36.0/27.4/52.6 | 35.1/27.6/48.2 | 33.8/25.8/49.1 |
| LLaVA NeXT Video (Zhang et al., 2024b) | 7B | 21.7/15.5/36.2 | 24.1/17.3/39.9 | 26.8/19.6/42.3 | 26.3/19.5/40.4 | 24.7/17.9/39.8 |
| InternVL2 (Chen et al., 2023) | 7B | 20.4/15.1/31.6 | 23.1/17.3/34.6 | 24.9/18.3/38.7 | 22.7/17.1/33.8 | 22.9/17.1/34.9 |
| InternVL2.5 (Chen et al., 2024) | 7B | 26.4/20.4/37.2 | 28.4/22.7/37.9 | 31.6/26.4/39.4 | 29.6/24.4/37.7 | 29.1/23.5/38.2 |
| InternVL2.5 (Chen et al., 2024) | 72B | 28.7/22.4/40.0 | 28.6/23.3/37.3 | 34.0/28.2/42.7 | 30.8/25.7/38.5 | 30.5/24.8/39.5 |
| MiniCPM-V 2.6 (Yao et al., 2024) | 7B | 28.9/19.7/53.6 | 29.4/21.0/48.8 | 32.0/23.7/49.3 | 32.2/23.3/52.1 | 30.5/21.9/50.5 |
| Tarsier (Wang et al., 2024a) | 7B | 30.0/22.2/45.9 | 30.0/22.6/44.4 | 33.4/24.9/50.7 | 31.2/23.9/45.1 | 31.1/23.4/46.5 |
| Qwen2-VL (Wang et al., 2024b) | 7B | 23.7/15.8/47.7 | 23.0/15.1/47.8 | 24.9/16.2/53.1 | 24.8/16.8/47.2 | 24.0/15.9/49.1 |
| Qwen2-VL (Wang et al., 2024b) | 72B | 24.5/16.3/49.4 | 22.5/14.7/47.8 | 24.6/15.8/56.3 | 26.5/17.4/55.7 | 24.2/15.8/51.9 |
| Qwen2.5-VL (Bai et al., 2025) | 7B | 32.8/25.3/46.6 | 32.7/25.9/44.2 | 34.8/27.6/47.3 | 34.0/27.7/44.1 | 33.5/26.5/45.5 |
| CARE$_{\text{stage-I}}$ | 7B | 32.1/22.6/55.3 | 31.3/22.2/53.1 | 33.2/23.2/58.4 | 33.6/23.8/57.1 | 32.4/22.9/55.7 |
| CARE | 7B | 30.9/21.1/57.2 | 31.5/21.9/55.6 | 31.8/21.3/62.6 | 32.6/23.0/55.8 | 31.7/21.8/57.8 |

## 4.2 STAGE-II: RETRIEVAL ADAPTATION

After Stage-I, CARE achieves precise alignment between pixel space and fine-grained text space. To shift the model output from the vocabulary space $\mathbb{R}^{D_v}$ to the embedding space $\mathbb{R}^{D_e}$, we use a similar method as Jiang et al. (2024b;a), employing an Explicit One-word Limitation (EOL) prompt to extract embeddings from CARE. Specifically, there are two steps: (1) given an EOL prompt: "`<sent> Summary of the above sentence in one word:`", the model is instructed to summarize the sentence $s_i$ in the next token; (2) we use the hidden states in the next token generation step as the final embeddings $f_i$. Then, we train the model on an NLI dataset (Gao et al., 2021) where each sample contains a sentence $s_i$, its positive $s_i^+$ and its hard negative $s_i^-$. Since there are no video inputs during Stage-II, we freeze the vision encoder and train the LLM only. Our training objective is given as:

$$\mathcal{L} = -\log \frac{e^{\cos\left(f_i, f_i^+\right)/\tau}}{\sum_{j=1}^{N}\left(e^{\cos\left(f_i, f_j^+\right)/\tau} + e^{\cos\left(f_i, f_j^-\right)/\tau}\right)}, \quad (4)$$

where $f_i$, $f_i^+$, $f_i^-$ denote the embeddings of the sentence $s_i$, its positive $s_i^+$ and its hard negative $s_i^-$, respectively. $\cos(\cdot)$ is the cosine similarity function. $\tau$ is the temperature hyperparameter.

## 5 EXPERIMENTS

In this section, we present the experiments on CAREBENCH. Section 5.1 shows the experiment settings. Section 5.2 and 5.3 analyze the results on video captioning and retrieval. In section 5.5, we conduct ablations to show the effectiveness of our methods. Additional experiments on other benchmarks and smaller MLLMs are included in Appendix B.

Table 4: **Video retrieval performance of some state-of-the-arts methods on CAREBENCH.** All the results are reported in zero-shot setting.

| Model | CAREBENCH General Retrieval | | | | | |
| --- | --- | --- | --- | --- | --- | --- |
| | Text-to-Video | | | Video-to-Text | | |
| | **R@1** | **R@5** | **R@10** | **R@1** | **R@5** | **R@10** |
| **CLIP-based Models** | | | | | | |
| CLIP B/16 (Radford et al., 2021) | 45.7 | 79.6 | 89.1 | 48.4 | 82.4 | 90.8 |
| CLIP L/14 (Radford et al., 2021) | 51.2 | 83.4 | 90.6 | 54.7 | 86.9 | 93.6 |
| LanguageBind (Zhu et al., 2024) | 64.3 | 91.0 | 96.3 | 59.5 | 88.0 | 95.0 |
| Long-CLIP B/14 (Zhang et al., 2024a) | 59.2 | 85.3 | 92.1 | 55.8 | 84.7 | 92.9 |
| Long-CLIP L/14 (Zhang et al., 2024a) | 62.7 | 88.8 | 95.7 | 60.3 | 88.8 | 94.9 |
| InternVideo2$_{stage2}$ 1B (Wang et al., 2024d) | 72.5 | 93.7 | 97.3 | 69.5 | 94.6 | 97.8 |
| **MLLMs** | | | | | | |
| LLaVA NeXT Video 7B (Zhang et al., 2024b) | 22.4 | 51.5 | 65.3 | 25.2 | 54.4 | 67.7 |
| MiniCPM-V 2.6 (Yao et al., 2024) | 8.2 | 26.9 | 38.4 | 16.7 | 39.9 | 55.8 |
| InternVL2 8B (Chen et al., 2023) | 34.6 | 67.1 | 80.2 | 35.1 | 68.5 | 82.0 |
| Tarsier 7B (Wang et al., 2024a) | 26.8 | 64.6 | 83.5 | 32.3 | 68.0 | 84.4 |
| Qwen2-VL 7B (Wang et al., 2024b) | 30.9 | 64.7 | 79.1 | 32.9 | 69.6 | 82.7 |
| **Contrastively trained MLLMs** | | | | | | |
| LLaVA NV 7B (Zhang et al., 2024b) | 66.9 | 89.4 | 96.0 | 62.7 | 89.2 | 95.4 |
| MiniCPM-V 2.6 (Yao et al., 2024) | 71.0 | 92.2 | 97.0 | 69.3 | 92.8 | 97.1 |
| InternVL2 8B (Chen et al., 2023) | 72.1 | 92.6 | 96.8 | 73.6 | 93.4 | 97.4 |
| Tarsier 7B (Wang et al., 2024a) | 71.0 | 93.8 | 97.8 | 70.6 | 94.2 | 98.0 |
| Qwen2-VL 7B (Wang et al., 2024b) | 76.6 | 95.3 | **98.7** | 77.4 | 95.6 | 98.7 |
| Qwen2.5-VL 7B (Bai et al., 2025) | **77.5** | 95.4 | 98.6 | 75.5 | 95.8 | 98.7 |
| **CARE** | 77.0 | **95.6** | **98.7** | 79.0 | **96.8** | **99.1** |

Table 5: **Spatiotemporal retrieval results of video retrieval on CAREBENCH.** LLaVA NV 7B is short for LLaVA NeXT Video 7B. We train all the MLLMs contrastively on NLI dataset to enable them to generate video embeddings. All the results are reported in zero-shot setting.

| Model | CAREBENCH Spatial Retrieval | | | | | | CAREBENCH Temporal Retrieval | | | | | | ReBias% |
| --- | --- | --- | --- | --- | --- | --- | --- | --- | --- | --- | --- | --- | --- |
| | Text-to-Video | | | Video-to-Text | | | Text-to-Video | | | Video-to-Text | | | |
| | R@1 | R@5 | R@10 | R@1 | R@5 | R@10 | R@1 | R@5 | R@10 | R@1 | R@5 | R@10 | |
| **CLIP-based Models** | | | | | | | | | | | | | |
| CLIP B/16 (Radford et al., 2021) | 45.6 | 79.0 | 89.2 | 47.6 | 80.9 | 90.8 | 30.3 | 65.1 | 79.8 | 35.8 | 71.0 | 85.8 | 17.75 |
| CLIP L/14 (Radford et al., 2021) | 49.0 | 81.9 | 91.4 | 55.4 | 85.6 | 93.0 | 33.5 | 70.3 | 84.0 | 39.7 | 76.2 | 87.9 | 16.52 |
| LanguageBind (Zhu et al., 2024) | 64.7 | 90.8 | 96.8 | 61.0 | 87.2 | 94.5 | 39.8 | 77.3 | 90.5 | 42.2 | 77.6 | 91.7 | 18.10 |
| Long-CLIP B/14 (Zhang et al., 2024a) | 62.5 | 86.0 | 92.7 | 53.8 | 84.1 | 92.7 | 32.0 | 65.4 | 79.3 | 29.7 | 67.3 | 84.1 | 31.88 |
| Long-CLIP L/14 (Zhang et al., 2024a) | 65.6 | 90.9 | 96.0 | 61.0 | 88.3 | 94.4 | 33.2 | 68.8 | 81.6 | 34.5 | 71.9 | 86.6 | 31.77 |
| InternVideo2$_{stage2}$ 1B (Wang et al., 2024d)[†] | 72.4 | 94.2 | 97.4 | 62.7 | 90.5 | 95.9 | 46.0 | 80.8 | 91.9 | 46.6 | 82.5 | 92.5 | 16.58 |
| **MLLMs** | | | | | | | | | | | | | |
| LLaVA NV 7B (Zhang et al., 2024b) | 34.1 | 63.1 | 76.0 | 31.1 | 63.7 | 75.1 | 18.6 | 48.1 | 62.4 | 20.7 | 47.1 | 62.4 | 32.32 |
| MiniCPM-V 2.6 (Yao et al., 2024) | 6.6 | 25.2 | 35.7 | 13.3 | 38.2 | 53.5 | 11.8 | 35.8 | 52.2 | 16.6 | 47.4 | 64.4 | 24.41 |
| InternVL2 8B (Chen et al., 2023) | 40.4 | 72.9 | 83.8 | 40.3 | 73.0 | 85.7 | 29.3 | 62.5 | 77.4 | 27.1 | 59.8 | 75.9 | 19.31 |
| Tarsier 7B (Wang et al., 2024a) | 40.5 | 74.0 | 88.1 | 41.9 | 75.0 | 87.4 | 26.8 | 64.6 | 83.5 | 32.3 | 68.0 | 84.4 | 13.15 |
| Qwen2-VL 7B (Wang et al., 2024b) | 28.1 | 61.3 | 76.1 | 31.6 | 65.6 | 80.4 | 24.3 | 61.5 | 78.4 | 26.4 | 59.2 | 76.1 | 5.28 |
| **Contrastively trained MLLMs** | | | | | | | | | | | | | |
| LLaVA NV 7B (Zhang et al., 2024b) | 68.0 | 92.0 | 96.2 | 65.0 | 90.0 | 95.9 | 43.3 | 76.9 | 88.9 | 40.1 | 75.4 | 88.7 | 22.69 |
| MiniCPM-V 2.6 (Yao et al., 2024) | 71.7 | 93.6 | 98.0 | 67.6 | 92.3 | 97.7 | 50.5 | 82.9 | 92.1 | 46.1 | 80.9 | 93.3 | 16.89 |
| InternVL2 8B (Chen et al., 2023) | 76.1 | 94.1 | 97.6 | 74.3 | 94.5 | 97.6 | 48.1 | 76.8 | 89.0 | 47.6 | 78.2 | 90.3 | 25.02 |
| Tarsier 7B (Wang et al., 2024a) | 70.2 | 94.0 | 98.2 | 67.4 | 93.5 | 97.4 | 50.1 | 84.1 | 92.8 | 50.0 | 84.7 | 94.9 | 14.04 |
| Qwen2-VL 7B (Wang et al., 2024b) | **78.2** | 95.5 | 98.5 | 75.4 | 95.0 | 98.1 | **51.9** | 84.8 | 94.9 | 52.7 | 85.4 | **95.2** | 16.30 |
| Qwen2.5-VL 7B (Bai et al., 2025) | 77.0 | 95.6 | **98.8** | 72.6 | 94.8 | 97.8 | 51.4 | 84.1 | 93.9 | 50.5 | 83.1 | 93.5 | 17.56 |
| **CARE** | 76.8 | **96.3** | 98.7 | **78.1** | **95.8** | **99.3** | 50.7 | **85.3** | 94.4 | **53.4** | **86.3** | 94.0 | 17.53 |

[†] InternVideo2$_{stage2}$ is tested without match header for fairness.

## 5.1 SETTINGS

In Stage-I, we train Qwen2-VL (Wang et al., 2024b) for about 400 GPU hours with a learning rate of 2e-5, batch size of 64, max pixel of 460,800, and 16 input frames. For Stage-II, CARE$_{stage-II}$ is initialized from Stage-I and trained on NLI dataset with the video backbone frozen. Due to text-only

contrastive learning, Stage-II only requires 24 GPU hours. We set epoch, batch size, and warmup ratio to 2, 768, and 0.2, respectively, and fully fine-tune $\text{CARE}_{\text{stage-II}}$ with learning rate of 2e-4.

## 5.2 Video captioning

In Table 2 and Table 3, we present quantitative comparison of the video captioning task on CAREBENCH between CARE and popular VLMs. Results are reported in zero-shot setting following our CapST metric. We use DeepSeek-V3 (DeepSeek-AI et al., 2024) to serve as the LLM judge. The number of input frames are set to 32. The default prompt is "`Describe the video in detail.`" unless the official research (Zhang et al., 2024b) recommends a specific one.

As illustrated in Table 2 and Table 3, our model has demonstrated superior performance across all the categories, surpassing all existing open-source models currently available. Considering the disparity between the models' parameters and their performance, even the most powerful MLLM, Qwen2-VL 72B, exhibits a significant performance gap when compared to our 7B CARE. This indicates that all current models still lack the ability to provide highly detailed, comprehensive, and fine-grained video descriptions. Additionally, it can be observed that whether the model has undergone stage II training does not affect its captioning performance. These promising results demonstrate that even a small-scale 7B model is capable of understanding the details within videos, including dynamic actions and static object elements and can have outstanding captioning and retrieval abilities simultaneously.

## 5.3 Video retrieval

We compare CLIP-based models, contrastively trained MLLMs and our CARE on CAREBENCH, following the setting of 32 input frames. Table 4 and Table 5 present the general retrieval performance and spatiotemporal retrieval performance on CAREBENCH. General retrieval uses first-stage annotations, while spatial and temporal retrieval leverage spatial captions and temporal captions from second-stage. All tasks employ Recall at Rank K (R@K, higher is better) in a zero-shot setting. The following observations can be concluded according to our analysis:

**MLLMs perform better than CLIP-based models on video retrieval.** CLIP-based models have long dominated retrieval performance benchmarks. However, as demonstrated in Table 4, MLLMs trained with contrastive learning exhibit significantly enhanced retrieval capabilities, surpassing their predecessors in performance. Our CARE yields the most favorable results, surpassing CLIP, Long-CLIP, LanguageBind, InternVideo2 and all the other MLLMs.

**VLMs have inherent biases in their spatiotemporal understanding and excel at leveraging spatial shortcuts for video understanding.** According to Table 5, all the models exhibit imbalance in spatiotemporal understanding, with spatial retrieval performance significantly outperforming temporal retrieval performance. When we switch from general retrieval to spatial retrieval, the performance drops for VLMs is small (*Avg. R@1:* Qwen2-VL –0.20, Tarsier –2.00, MiniCPM-V 2.6 –0.50). In contrast, the drop is substantial for Temporal Retrieval (*Avg. R@1:* Qwen2-VL –24.70, Tarsier –20.75, MiniCPM-V 2.6 –21.85). Even when most action-related cues are removed from captions, VLMs still maintain comparable performance. This indicates that these VLMs rely on scene cues as a shortcut rather than using the detailed action information when they describe videos. Such a bias highlights the need for improved methods to enhance temporal understanding capabilities in video understanding tasks.

## 5.4 Video QA

To demonstrate CARE's generalization capability on out-of-domain tasks, we conduct video QA evaluation on MVBench (Li et al., 2024) and TVBench (Cores et al., 2024). Results for models other than CARE are from the official reports of benchmarks. The results in Table 6 show that CARE demonstrates strong generalization on general video understanding benchmarks.

## 5.5 Ablation study

In this section, we conduct experiments to further investigate the effect of our proposed two-stage SFT. Using the same setting as mentioned in Section 5.1 and building upon the Qwen2-VL model (Wang et al., 2024b), we perform a quantitative analysis to evaluate the impact of different

Table 6: **Performance of Video QA on MVBench (Li et al., 2024) and TVBench (Cores et al., 2024).** Results for models other than CARE are from the official reports of benchmarks.

| Models | **MVBench** (Li et al., 2024) | **TVBench** (Cores et al., 2024) |
|---|---|---|
| GPT-4o | 47.8 | 35.8 |
| VideoChat2 | 51.0 | 33.0 |
| PLLaVA-7B | 46.6 | 34.2 |
| Gemini 1.5 Pro | **60.5** | *46.5* |
| CARE | *60.4* | **50.1** |

Table 7: **Effect of the two-stage training.** Four model settings are included: the baseline, CARE with fine-grained caption adaptation only, CARE with retrieval adaptation only, and CARE with full two-stage SFT. The evaluation metrics include Avg. R@1, which denotes the average text-to-video and video-to-text R@1 on CAREBENCH General Retrieval, and Avg. F1, which represents the average action/object F1 on CAREBENCH. Unified Score is the average of R@1 and F1.

| Setting | Retrieval
Avg. R@1 | Caption
Avg. F1 | Overall
Unified Score |
|---|---|---|---|
| Baseline | 25.6 | 26.8 | 26.2 |
| +Fine-Grained Caption Adaptation | 17.6(-8.0) | 33.8(+7.0) | 25.7(-0.5) |
| +Retrieval Adaptation | 77.0(+51.4) | 28.2(+1.4) | 52.6(+26.4) |
| +Fine-Grained Caption Adaptation & Retrieval Adaptation | 78.0(+52.4) | 33.4(+6.6) | 55.7(+29.5) |

stages on the model's performance in video captioning and retrieval tasks, as shown in Table 7. Our baseline model, Qwen2-VL (Wang et al., 2024b), shows strong captioning skills (Avg. F1 26.8) but struggles with retrieval tasks (Avg. R@1 25.6) without retrieval adaptation. Adding fine-grained caption adaptation greatly improves the model's captioning ability (Avg. F1 +7.0) at a slight cost to retrieval performance (Avg. R@1 -8.0). On the other hand, using only retrieval adaptation gives the model excellent retrieval capabilities (Avg. R@1 +51.4), which is a big improvement over the baseline. After both training stages, our model not only excels in detailed video description but also achieves top-level retrieval performance. Interestingly, we have uncovered evidence that video retrieval and captioning tasks can mutually enhance each other: retrieval adaptation improves the baseline's video captioning performance by **+1.4** (Avg. F1 26.8 → 28.2), and the high-quality fine-tuning of fine-grained caption adaptation further boosts the retrieval adapted model by **+1** (Avg. R@1 77.0 → 78.0).

## 6 CONCLUSION

In this work, we present CAREBENCH, a fine-grained benchmark for video captioning and retrieval, featuring 1,000 videos with high-quality human-annotated descriptions. Each caption is structured hierarchically to cover four key aspects: overall summary, static object descriptions, dynamic action descriptions, and miscellaneous details such as filming styles. We also propose ReBias and CapST, two novel metrics for assessing retrieval and captioning performance. Additionally, we develop CARE, a unified baseline for both tasks, leveraging a two-stage supervised fine-tuning approach to generate detailed captions and extract video features. Experiments show that CARE outperforms specialized models in both fine-grained retrieval and captioning. Our work highlights the potential of unifying video captioning and retrieval tasks under a single framework, challenging the traditional methods. However, our model doesn't address problems about VLMs' spatiotemporal bias. Look ahead, future research could explore further integration of both tasks and try to develop a more balanced model.

## ACKNOWLEDGEMENTS

This work is supported by the Basic Research Program of Jiangsu (No. BK20250009) and the Collaborative Innovation Center of Novel Software Technology and Industrialization.

## REPRODUCIBILITY STATEMENT

We provide comprehensive materials to reproduce our results. Model and training details (including loss functions, hyperparameters, and training settings) are in Section 4 and 5.1. Dataset sources, licenses, and annotation steps are documented in Section 3 and Appendix H.

## ETHICAL STATEMENT

(1) *Human annotators.* We pay human annotators above the legally mandated minimum wage in accordance with the laws where the research is conducted.

(2) *Biases in benchmark annotations.* The authors are aware of the potential for bias in the annotations of our benchmark. These annotations may inadvertently reflect the annotators' perspectives and biases. We have tried to minimize the bias during the expert refinement and each annotation is cross-checked by two human experts.

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

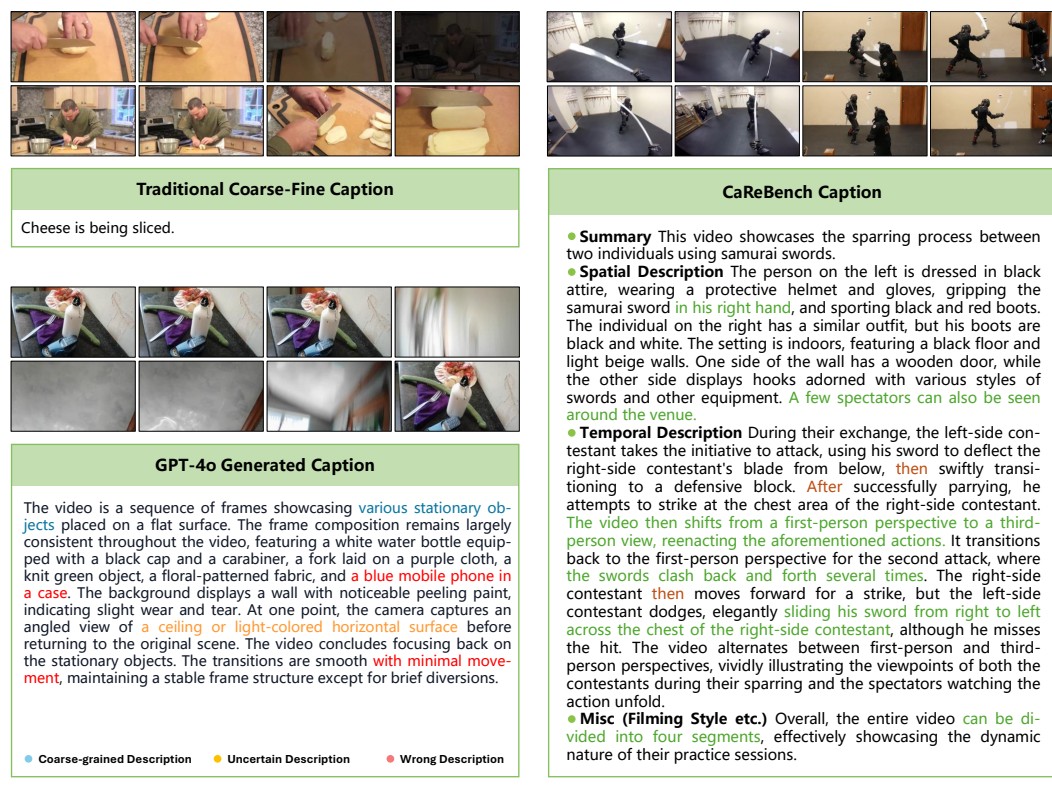

Figure 4: **Comparision of captions between MSR-VTT(Xu et al., 2016), GPT-4o generated data(Cui et al., 2024) and CAREBENCH.** The caption in the upper left corner is from MSR-VTT(Xu et al., 2016). It only contains short-text coarse descriptions. The annotation located in the lower left corner is generated by GPT-4o sourced from ShareGPT-4o(Cui et al., 2024). It has some coarse-grained, uncertain and wrong descriptions. The fine-grained caption on the right is selected from CAREBENCH and is created by our human annotator following the pipeline. The green sentences are fine-grained descriptions and the brown words show the temporal sequences in the video.

# A    CASE STUDY

Benchmarks like MSRVTT (Xu et al., 2016) rely on brief short captions. As shown in Figure 4, the MSRVTT caption in the upper-left corner overlooks key details, such as the contents of the kitchen and the attire of the man. Captions annotated by LLMs may have coarse-grained, uncertain and wrong descriptions. As shown in Figure 4, GPT-4o erroneously identifies the slipper beneath the phone as a phone case and describes the camera's violent shaking as "minimal movement." The fine-grained caption on the right is selected from CAREBENCH and is created by human. The green sentences are fine-grained descriptions and the brown words show the action sequences in the video. For more sample of CAREBENCH, see the end of the appendix.

# B    ADDITIONAL EXPERIMENTS

## B.1    EXPERIMENTS ON TRADITIONAL BENCHMARKS

We compare CLIP-based models, MLLMs, and CARE on traditional retrieval benchmarks. All the experiments follow the setting of 32 input frames. Table 8 and Table 9 present the retrieval performance of all the models on MSR-VTT (Xu et al., 2016), MSVD (Chen & Dolan, 2011) and DiDeMo (Hendricks et al., 2017). All the retrieval results are reported in zero-shot setting. Table 10 illustrates the captioning performance of popular models on DREAM-1K (Wang et al., 2024a).

Table 8: **Video retrieval performance on MSR-VTT (Xu et al., 2016) and MSVD (Chen & Dolan, 2011).**

| Model | MSR-VTT (Xu et al., 2016) | | | | | | MSVD (Chen & Dolan, 2011) | | | | | |
| | Text-to-Video | | | Video-to-Text | | | Text-to-Video | | | Video-to-Text | | |
| | R@1 | R@5 | R@10 | R@1 | R@5 | R@10 | R@1 | R@5 | R@10 | R@1 | R@5 | R@10 |
|---|---|---|---|---|---|---|---|---|---|---|---|---|
| **CLIP-based Models** | | | | | | | | | | | | |
| CLIP B/16 (Radford et al., 2021) | 33.8 | 56.1 | 66.6 | 30.5 | 53.8 | 65.5 | 37.0 | 64.2 | 74.1 | 60.5 | 79.9 | 87.5 |
| CLIP L/14 (Radford et al., 2021) | 36.7 | 58.8 | 68.0 | 32.8 | 54.7 | 66.2 | 41.1 | 68.8 | 77.5 | 68.1 | 85.5 | 91.8 |
| LanguageBind (Zhu et al., 2024) | 42.1 | 65.9 | 75.5 | 40.1 | 65.4 | 73.9 | 50.0 | 77.7 | 85.6 | 75.1 | 90 | 94.2 |
| Long-CLIP B/14 (Zhang et al., 2024a) | 38.7 | 62.3 | 70.6 | 34.4 | 57.7 | 68.2 | 40.4 | 68.0 | 77.7 | 63.4 | 81.6 | 87.8 |
| Long-CLIP L/14 (Zhang et al., 2024a) | 40.9 | 65.5 | 74.6 | 36.2 | 62.2 | 71.5 | 46.5 | 73.5 | 82.0 | 69.3 | 86.0 | 90.3 |
| InternVideo2$_{stage2}$ 1B (Wang et al., 2024d)[†] | 44.2 | 70.1 | 78.1 | 40.5 | 66.9 | 76.3 | 53.0 | 79.1 | 87.2 | 74.6 | 88.5 | 93.4 |
| **Contrastively Trained MLLMs** | | | | | | | | | | | | |
| LLaVA NeXT Video 7B (Zhang et al., 2024b) | 40.3 | 64.9 | 74.1 | 30.5 | 58.0 | 69.0 | 47.3 | 75.7 | 83.7 | 51.9 | 74.3 | 81.8 |
| InternVL2 8B (Chen et al., 2023) | 44.6 | 69.3 | 77.4 | 40.8 | 66.6 | 76.5 | 47.7 | 75.9 | 83.9 | 64.2 | 81.3 | 87.2 |
| MiniCPM-V 2.6 (Yao et al., 2024) | 44.7 | 69.7 | 77.8 | 41.6 | 68.7 | 77.6 | 50.5 | 78.7 | 85.8 | 69.1 | 84.6 | 90.2 |
| Tarsier 7B (Wang et al., 2024a) | 43.4 | 69.2 | 77.0 | 35.8 | 62.5 | 72.3 | 52.1 | 79.7 | 86.5 | 67.8 | 88.8 | 93.1 |
| Qwen2-VL 7B (Wang et al., 2024a) | 46.9 | 69.2 | 79.7 | 43.4 | 69.2 | 78.8 | 53.3 | 79.7 | 86.5 | 73.7 | 89.6 | 92.4 |
| **CARE** | 43.9 | 67.0 | 75.7 | 41.7 | 68.1 | 76.2 | 52.6 | 79.2 | 86.6 | 74.6 | 87.9 | 92.4 |

[†] InternVideo2$_{stage2}$ is tested without match header for fairness.

Table 9: **Video retrieval performance on DiDeMo (Hendricks et al., 2017).**

| Model | DiDeMo | | | | | |
| | Text-to-Video | | | Video-to-Text | | |
| | **R@1** | **R@5** | **R@10** | **R@1** | **R@5** | **R@10** |
|---|---|---|---|---|---|---|
| **CLIP-based Models** | | | | | | |
| CLIP B/16 (Radford et al., 2021) | 23.5 | 46.3 | 55.2 | 22.2 | 43.8 | 54.0 |
| CLIP L/14 (Radford et al., 2021) | 24.1 | 48.0 | 58.2 | 23.8 | 44.9 | 54.0 |
| LanguageBind (Zhu et al., 2024) | 35.6 | 63.6 | 71.7 | 35.6 | 62.8 | 71.8 |
| Long-CLIP B/14 (Zhang et al., 2024a) | 30.3 | 52.4 | 63.7 | 24.8 | 52.8 | 63.4 |
| Long-CLIP L/14 (Zhang et al., 2024a) | 32.4 | 56.2 | 65.2 | 28.5 | 54.1 | 64.7 |
| InternVideo2$_{stage2}$ 1B (Wang et al., 2024d)[†] | 35.0 | 63.7 | 74.1 | 35.5 | 60.7 | 70.7 |
| **Contrastively Trained MLLMs** | | | | | | |
| LLaVA NeXT Video 7B (Zhang et al., 2024b) | 36.0 | 62.3 | 71.7 | 31.4 | 58.0 | 68.0 |
| InternVL2 8B (Chen et al., 2023) | 39.7 | 65.6 | 74.1 | 35.5 | 64.0 | 72.2 |
| MiniCPM-V 2.6 (Yao et al., 2024) | 40.6 | 65.2 | 74.2 | 35.7 | 61.6 | 70.1 |
| Tarsier 7B (Wang et al., 2024a) | 42.1 | 68.2 | 77.1 | 39.5 | 64.6 | 73.7 |
| Qwen2-VL 7B (Wang et al., 2024a) | 46.1 | 69.6 | 77.6 | 42.1 | 66.1 | 76.3 |
| **CARE** | 41.4 | 68.5 | 77.1 | 39.1 | 66.0 | 75.8 |

[†] InternVideo2$_{stage2}$ is tested without match header for fairness.

## B.2 EXPERIMENTS ON SMALLER MLLMS

We additionally benchmark small contrastively-trained MLLMs (1B/2B) on CAREBENCH and MSR-VTT (Xu et al., 2016). The training follows the setting of Stage-II and consumes only 2.26 GPU hours and 6.4 GPU hours for 1B and 2B models, respectively. Table 11 and Table 12 report Recall@{1,5,10} for both text-to-video and video-to-text retrieval. As shown, competitive InternVL 2.5 1B/2B surpass Long-CLIP and narrow much of the gap to earlier 7B MLLMs—highlighting the importance of training objectives and data over parameter count alone—while our 7B model still achieves the strongest overall results. All methods follow the same input preprocessing and evaluation settings to ensure comparability.

Table 10: **Video caption performance of popular models on DREAM-1K (Wang et al., 2024a).** We report F1/Recall/Precision for each category. # Params denotes the number of LLM parameters.

| Model | # Params | DREAM-1K | | | | | |
| --- | --- | --- | --- | --- | --- | --- | --- |
| | | Live-Action Movies | Animation Movies | Stock Videos | YouTube Videos | TikTok-Style Short | Overall |
| GPT-4o mini | - | 34.5/32.7/36.6 | 28.9/26.0/32.6 | 37.9/38.0/37.9 | 33.5/30.2/37.5 | 34.7/29.3/42.4 | 34.0/31.2/37.4 |
| LLaVA NeXT Video (Zhang et al., 2024b) | 7B | - | - | - | - | - | - |
| InternVL2 (Chen et al., 2023) | 7B | 27.3/27.1/27.4 | 20.6/18.1/23.8 | 33.3/33.0/33.5 | 26.9/24.2/30.2 | 25.7/21.2/32.7 | 26.9/24.7/29.5 |
| InternVL2.5 (Chen et al., 2024) | 7B | - | - | - | - | - | - |
| InternVL2.5 (Chen et al., 2024) | 72B | - | - | - | - | - | - |
| MiniCPM-V 2.6 (Yao et al., 2024) | 7B | 30.5/27.7/33.8 | 24.8/22.5/27.8 | 35.4/35.0/35.8 | 29.5/28.0/31.3 | 31.6/26.5/38.9 | 30.5/27.9/33.5 |
| Tarsier (Wang et al., 2024a) | 7B | 36.6/34.8/38.5 | 29.3/25.5/34.6 | 39.6/35.5/44.7 | 33.0/28.4/39.2 | 33.6/26.9/44.6 | 34.6/30.2/40.3 |
| Qwen2-VL (Wang et al., 2024b) | 7B | 27.7/24.2/32.5 | 22.2/18.4/28.0 | 37.0/38.0/36.1 | 30.7/27.0/35.5 | 29.1/23.8/37.6 | 29.6/26.3/33.9 |
| Qwen2-VL (Wang et al., 2024b) | 72B | 32.1/30.6/33.7 | 27.6/23.9/32.6 | 41.1/41.1/41.2 | 32.0/27.7/38.1 | 32.1/26.4/41.0 | 33.2/29.9/37.3 |
| CARE$_{stage-1}$ | 7B | 40.8/41.9/39.7 | 33.7/31.6/36.0 | 44.0/45.2/43.0 | 34.5/32.5/36.6 | 38.4/33.7/44.7 | 38.4/37.0/40.0 |
| CARE | 7B | 41.9/42.1/41.8 | 32.0/30.0/34.2 | 44.2/45.6/42.8 | 34.5/32.3/37.1 | 37.3/33.7/41.7 | 38.1/36.8/39.5 |

Table 11: **Video retrieval performance of small MLLMs on CAREBENCH.** All the results are reported in zero-shot setting.

| Model | CAREBENCH General Retrieval | | | | | |
| --- | --- | --- | --- | --- | --- | --- |
| | Text-to-Video | | | Video-to-Text | | |
| | R@1 | R@5 | R@10 | R@1 | R@5 | R@10 |
| **CLIP-based Models** | | | | | | |
| CLIP B/16 (Radford et al., 2021) | 45.7 | 79.6 | 89.1 | 48.4 | 82.4 | 90.8 |
| CLIP L/14 (Radford et al., 2021) | 51.2 | 83.4 | 90.6 | 54.7 | 86.9 | 93.6 |
| LanguageBind (Zhu et al., 2024) | 64.3 | 91.0 | 96.3 | 59.5 | 88.0 | 95.0 |
| Long-CLIP B/14 (Zhang et al., 2024a) | 59.2 | 85.3 | 92.1 | 55.8 | 84.7 | 92.9 |
| Long-CLIP L/14 (Zhang et al., 2024a) | 62.7 | 88.8 | 95.7 | 60.3 | 88.8 | 94.9 |
| InternVideo2$_{stage2}$ 1B (Wang et al., 2024d) | 72.5 | 93.7 | 97.3 | 69.5 | 94.6 | 97.8 |
| **Contrastively trained MLLMs** | | | | | | |
| LLaVA NV 7B (Zhang et al., 2024b) | 66.9 | 89.4 | 96.0 | 62.7 | 89.2 | 95.4 |
| MiniCPM-V 2.6 (Yao et al., 2024) | 71.0 | 92.2 | 97.0 | 69.3 | 92.8 | 97.1 |
| **InternVL2.5 1B (Chen et al., 2024)** | 66.3 | 92.6 | 97.3 | 63.6 | 90.2 | 96.1 |
| InternVL2 8B (Chen et al., 2023) | 72.1 | 92.6 | 96.8 | 73.6 | 93.4 | 97.4 |
| **InternVL2.5 2B (Chen et al., 2024)** | 73.4 | 93.9 | 97.9 | 73.4 | 92.9 | 97.4 |
| Tarsier 7B (Wang et al., 2024a) | 71.0 | 93.8 | 97.8 | 70.6 | 94.2 | 98.0 |
| Qwen2-VL 7B (Wang et al., 2024b) | 76.6 | 95.3 | **98.7** | 77.4 | 95.6 | 98.7 |
| **CARE** | **77.0** | **95.6** | **98.7** | **79.0** | **96.8** | **99.1** |

## C  QUANTITATIVE AND HUMAN-ALIGNED VALIDATION ON METRICS

We conduct additional quantitative and human-aligned validation to show that CapST reflects fine-grained caption quality, correlates with human judgment compared to n-gram metrics and remains stable across different LLM judges.

### C.1  N-GRAM METRICS ON CAREBENCH

N-gram metrics on CAREBENCH including BLEU@4, METEOR, ROUGE-L and CIDEr are shown in Table 16. Due to the extremely high richness of the vocabulary in fine-grained captions, sentences with similar semantics can differ greatly at the token level. Consequently, n-gram–based metrics are already close to zero and no longer meaningful.

### C.2  HUMAN-ALIGNED VALIDATION ON CAPST METRIC

To provide human-aligned validation on CapST, we invite 10 human experts and follow the Elo settings in Table 17 to perform "which-is-better" evaluations on CAREBENCH and show the Elo scores in Table 18.

Table 12: **Video retrieval performance of small MLLMs on MSR-VTT (Xu et al., 2016).** All the results are reported in zero-shot setting.

| Model | MSR-VTT | | | | | |
| | Text-to-Video | | | Video-to-Text | | |
| | R@1 | R@5 | R@10 | R@1 | R@5 | R@10 |
|---|---|---|---|---|---|---|
| **CLIP-based Models** | | | | | | |
| CLIP B/16 (Radford et al., 2021) | 33.8 | 56.1 | 66.6 | 30.5 | 53.8 | 65.5 |
| CLIP L/14 (Radford et al., 2021) | 36.7 | 58.8 | 68.0 | 32.8 | 54.7 | 66.2 |
| LanguageBind (Zhu et al., 2024) | 42.1 | 65.9 | 75.5 | 40.1 | 65.4 | 73.9 |
| Long-CLIP B/14 (Zhang et al., 2024a) | 38.7 | 62.3 | 70.6 | 34.4 | 57.7 | 68.2 |
| Long-CLIP L/14 (Zhang et al., 2024a) | 40.9 | 65.5 | 74.6 | 36.2 | 62.2 | 71.5 |
| InternVideo2$_{stage2}$ 1B (Wang et al., 2024d)[†] | 44.2 | 70.1 | 78.1 | 40.5 | 66.9 | 76.3 |
| **Contrastively trained MLLMs** | | | | | | |
| LLaVA NeXT Video 7B (Zhang et al., 2024b) | 40.3 | 64.9 | 74.1 | 30.5 | 58.0 | 69.0 |
| **InternVL2.5 1B (Chen et al., 2024)** | 41.3 | 64.4 | 73.8 | 36.3 | 61.6 | 70.9 |
| InternVL2 8B (Chen et al., 2023) | 44.6 | 69.3 | 77.4 | 40.8 | 66.6 | 76.5 |
| **InternVL2.5 2B (Chen et al., 2024)** | 41.9 | 68.4 | 75.7 | 39.7 | 65.8 | 75.5 |
| MiniCPM-V 2.6 (Yao et al., 2024) | 44.7 | 69.7 | 77.8 | 41.6 | 68.7 | 77.6 |
| Tarsier 7B (Wang et al., 2024a) | 43.4 | 69.2 | 77.0 | 35.8 | 62.5 | 72.3 |
| Qwen2-VL 7B (Wang et al., 2024a) | 46.9 | 69.2 | 79.7 | 43.4 | 69.2 | 78.8 |
| CARE | 43.9 | 67.0 | 75.7 | 41.7 | 68.1 | 76.2 |

Table 19 shows the pearson correlation coefficients between these metrics and Elo scores (e.i. human preference). The results indicate that: **(1)** All the n-gram-based metrics are significantly uncorrelated with the Elo score. **(2)** Action F1 and Object F1 demonstrate significant correlations with the Elo score, suggesting they better capture human preferences.

## C.3 LLM JUDGE STABILITY OF CAPST METRIC

To show the stability of CapST across different LLM judges, we evaluated all the models using an identical prompt on Deepseek V3 (DeepSeek-AI et al., 2024) (our implementation), Gemini 2.5 Pro Comanici et al. (2025) and GPT 4.1 OpenAI (2023).

For a intuitive comparison, we computed the correlations between these LLM judges. All inter-judge correlations exceed 0.96, indicating that CapST remains highly stable across different judges. The results are shown in Table 13, Table 14, Table 15.

Table 13: **Video caption performance of popular models on CAREBENCH using GPT 4.1 (OpenAI, 2023) as LLM judge.** Only overall scores are reported.

| Models | # Params | Action F1 | Action R | Action P | Object F1 | Object R | Object P |
|---|---|---|---|---|---|---|---|
| GPT-4o mini | - | 36.1 | 30.5 | 44.2 | 34.0 | 26.7 | 46.6 |
| LLaVA NeXT Video | 7B | 25.9 | 20.2 | 36.2 | 24.5 | 18.1 | 37.6 |
| InternVL2 | 7B | 23.9 | 21.4 | 27.2 | 23.5 | 18.3 | 32.7 |
| InternVL 2.5 | 7B | 27.3 | 21.0 | 38.8 | 30.1 | **24.9** | 37.9 |
| InternVL 2.5 | 72B | 30.0 | 22.9 | 43.6 | 31.6 | **26.3** | 39.5 |
| MiniCPM-V 2.6 | 7B | 32.7 | 25.0 | 47.1 | 30.6 | 22.4 | 48.0 |
| Tarsier | 7B | 27.2 | 19.0 | 48.1 | 32.0 | 24.2 | 47.4 |
| Qwen2-VL | 7B | 28.5 | 24.6 | 33.9 | 22.9 | 15.8 | 41.7 |
| Qwen2-VL | 72B | 29.9 | 23.0 | 42.7 | 23.2 | 15.6 | 45.5 |
| **CARE$_{stage-I}$** | 7B | **35.3** | **26.9** | **51.0** | 32.5 | 23.0 | **53.2** |
| **CARE** | 7B | **34.4** | **26.1** | **50.3** | 32.3 | 23.0 | **54.2** |

Table 14: **Video caption performance of popular models on CAREBENCH using Gemini 2.5 Pro (Comanici et al., 2025) as LLM judge.** Only overall scores are reported.

| Models | # Params | Action F1 | Action R | Action P | Object F1 | Object R | Object P |
|---|---|---|---|---|---|---|---|
| GPT-4o mini | - | 41.5 | 31.7 | 59.9 | 39.9 | 30.7 | 57.0 |
| LLaVA NeXT Video | 7B | 29.0 | 19.9 | 53.2 | 30.1 | 21.5 | 50.1 |
| InternVL2 | 7B | 28.6 | 23.1 | 37.4 | 28.5 | 21.4 | 42.8 |
| InternVL 2.5 | 7B | 30.4 | 21.1 | 53.9 | 35.6 | 28.7 | 47.1 |
| InternVL 2.5 | 72B | 33.4 | 23.4 | 58.4 | 36.8 | **29.5** | 48.9 |
| MiniCPM-V 2.6 | 7B | 35.3 | 24.7 | 62.2 | 36.4 | 26.1 | 60.2 |
| Tarsier | 7B | 32.1 | 21.5 | **63.6** | 36.4 | 27.1 | 55.4 |
| Qwen2-VL | 7B | 32.8 | 25.1 | 44.0 | 27.7 | 18.4 | 56.2 |
| Qwen2-VL | 72B | 34.8 | 26.1 | 52.1 | 28.1 | 18.4 | 60.0 |
| **CARE**$_{stage-I}$ | 7B | **41.2** | **31.5** | 59.6 | **37.1** | 26.0 | **64.8** |
| **CARE** | 7B | **40.3** | **30.2** | 60.7 | 36.4 | 25.1 | **66.6** |

Table 15: **Inter-judge consistency analysis via Pearson correlation.** We report the Pearson correlation coefficients (and corresponding p-values) between evaluation results produced by DeepSeek V3 (DeepSeek-AI et al., 2024), GPT 4.1 (OpenAI, 2023), and Gemini 2.5 Pro (Comanici et al., 2025).

| **DeepSeek-V3 vs GPT-4.1** | | | **DeepSeek-V3 vs Gemini-2.5-Pro** | | | **GPT-4.1 vs Gemini-2.5-Pro** | | |
|---|---|---|---|---|---|---|---|---|
| Metric | Corr. | P-val | Metric | Corr. | P-val | Metric | Corr. | P-val |
| Action F1 | 0.97 | 5.6e-07 | Action F1 | 0.98 | 1.5e-07 | Action F1 | 0.98 | 3.5e-07 |
| Action R | 0.96 | 3.6e-06 | Action R | 0.97 | 5.4e-07 | Action R | 0.92 | 7.7e-05 |
| Action P | 0.94 | 1.8e-05 | Action P | 0.95 | 7.9e-06 | Action P | 0.92 | 4.9e-05 |
| Object F1 | 0.99 | 1.3e-08 | Object F1 | 0.98 | 1.5e-07 | Object F1 | 0.99 | 3.5e-09 |
| Object R | 0.99 | 2.4e-09 | Object R | 0.99 | 2.4e-08 | Object R | 0.99 | 1.2e-09 |
| Object P | 0.96 | 5.6e-06 | Object P | 0.99 | 3.1e-10 | Object P | 0.96 | 2.7e-06 |
| Avg | 0.97 | - | Avg | 0.98 | - | Avg | 0.96 | - |

# D LIMITATIONS

Although CARE demonstrates strong generalization ability on fine-grained video retrieval and captioning tasks, it exhibits a certain degree of performance drop compared to the baseline (i.e constrastively trained Qwen2-VL) on coarse-grained, traditional datasets such as MSR-VTT (Xu et al., 2016), as shown in Table 8 and Table 9. To further investigate this phenomenon, we provide in this section the results of CARE versus contrastively trained Qwen2-VL (Wang et al., 2024b) on MSVD (Chen & Dolan, 2011), MSR-VTT (Xu et al., 2016), DiDeMo (Hendricks et al., 2017), VDC (Chai et al., 2024) short captions, DREAM-1K (Wang et al., 2024a), ShareGPT-4o (Cui et al., 2024), and CAREBENCH, together with an explicit, quantitative analysis of how performance varies with caption length and action–object complexity across benchmarks.

## D.1 PERFORMANCE OF CARE AND QWEN2-VL 7B ACROSS DIFFERENT BENCHMARKS

Table 20 shows the results of CARE and contrastively trained Qwen2-VL on difference benchmarks. To ensure a fair comparison of model performance across datasets and to eliminate the influence of absolute *Avg Recall* values, we use the relative gap (percentage) in Equation (5). The results of performance gaps across difference benchmarks are illustrated in Table 21.

$$G = \frac{R_{\text{CARE}} - R_{Qwen}}{R_{Qwen}} \tag{5}$$

where $R_{\text{CARE}}$ is *Avg Recall* of CARE and $R_{Qwen}$ is *Avg Recall* of contrastively trained Qwen2-VL.

Table 16: **CAREBENCH results with BLEU@4, METEOR, ROUGE-L, and CIDEr.**

| Models | BLEU@4 | METEOR | ROUGE-L | CIDEr |
|---|---|---|---|---|
| LLaVA Video 7B (Zhang et al., 2024b) | 0.030 | 0.155 | 0.185 | 0.019 |
| MiniCPM-V 2.6 (Yao et al., 2024) | 0.022 | 0.139 | 0.163 | 0.025 |
| InternVL 2.5 8B (Chen et al., 2024) | 0.015 | 0.120 | 0.156 | 0.000 |
| Tarsier 7B (Wang et al., 2024a) | 0.012 | 0.093 | 0.151 | 0.000 |
| Qwen2-VL 7B (Wang et al., 2024b) | 0.013 | 0.114 | 0.128 | 0.009 |
| Qwen2-VL 72B (Wang et al., 2024b) | 0.018 | 0.115 | 0.167 | 0.000 |
| CARE | 0.012 | 0.100 | 0.153 | 0.000 |

Table 17: **Elo setting.**

| Parameter | Number |
|---|---|
| initial Elo mean | 1,000 |
| Elo standard deviation | 300 |
| base of logarithm | 10 |
| scaling factor | 400 |
| K-factor | 32 |
| minimum Elo rating | 700 |

### D.2 PEARSON CORRELATION ANALYSIS BETWEEN PERFORMANCE GAP AND BENCHMARK STATISTICS

To visualize how the relative performance gap numerically relates to each benchmark statistic, we performed a Pearson correlation analysis; the results are shown in Table 22.

According to the results illustrated above, we can come to the conclusions that:

1. CARE has a clear performance advantage on long-text, fine-grained tasks – especially those containing a large number of actions and objects. Pearson correlation analysis shows that the Relative Gap between our model and Qwen2-VL is strongly and positively correlated with Avg. Words and Avg. Objects, and moderately positively correlated with Avg. Actions. The more detailed the benchmark and the more objects and actions it contains, the more pronounced CARE's advantages become.

2. CARE underperforms Qwen2-VL on coarse-grained retrieval benchmarks with simple objects and actions within the 7-to-60-word range. In contrast to Qwen2-VL, CARE incorporates an additional Fine-grained Alignment SFT. Although more ablation studies are needed to verify how the SFT design affects performance, we can tentatively conclude that this extra training reduces the model's generalization on coarse-grained tasks.

## E OBJECT LEAKAGE IN TEMPORAL CAPTION

We've tried our best to reduce the object leakage (i.e. having unnecessary objects) in temporal caption. We here provide statistics and examples illustrating the degree of static/object leakage in temporal annotations. We extract the objects in spatial captions and temporal captions using Deepseek V3. The avg. objects in spatial caption is 10.76 and avg. objects in temporal caption is 6.82. The following shows two examples.

**Example 1:**

```
[video] v_00000044_3.mp4
[category] Personal Care
[subcategory] apply_eyebrows

[spatial_objects]
```

Table 18: **Elo scores of popular models on CAREBENCH.**

| Model | Elo Score |
|---|---|
| InternVL2 8B (Chen et al., 2023) | 889.31 |
| LLaVA NeXT Video 7B (Zhang et al., 2024b) | 914.27 |
| Qwen2-VL 7B (Wang et al., 2024b) | 959.34 |
| MiniCPM-V 2.6 (Yao et al., 2024) | 1051.14 |
| Tarsier 7B (Wang et al., 2024a) | 1061.59 |
| CARE | 1124.34 |

Table 19: **Pearson correlation coefficients between metrics and Elo scores.**

| Metrics | Elo Score |
|---|---|
| BLEU@4 | -0.41 |
| METEOR | -0.54 |
| ROUGE-L | -0.19 |
| CIDEr | -0.14 |
| Action F1 | 0.81 |
| Object F1 | 0.71 |

```
a woman with long black hair
a right eyebrow with brow dye
soft orange eyeshadow with silver glitter
pale pink lips
a pink wall with a black grid pattern
an upper garment with a blue base
an upper garment with white floral patterns
a small bottle of yellowish-brown brow dye
a small white mirror
a mirror with a yellow-edged border
a mirror with a silver design in the lower right corner
a brush with a black bristle head
a brush with a yellowish-brown stick
a brush with silver-white text branding.

[temporal_objects]
a person
a brow gel
a right eyebrow
a mirror
a small brush
```

**Example 2:**

```
[video] v_00007656_1.mp4
[category] Sports, Excercise
[subcategory] drop_golf

[spatial_objects]
a boy in a light blue short-sleeve shirt
a boy in khaki shorts
a boy wearing a white baseball cap
a boy wearing white-brown athletic shoes
a black golf club
a boy in a red short-sleeve shirt
a boy in khaki-striped shorts
```

Table 20: **Performance of CaRe and Qwen2-VL 7B across different datasets.**

| Model | ShareGPT-4o | CaReBench | MSVD | VDC short captions | MSR-VTT | DiDeMo | DREAM-1K |
|---|---|---|---|---|---|---|---|
| CaRe | 87.38 | 91.03 | 78.88 | 80.19 | 62.10 | 61.32 | 92.93 |
| Qwen2-VL 7B | 87.03 | 90.38 | 79.20 | 81.00 | 64.53 | 62.97 | 94.65 |

[†] Metrics are computed as the average of T2V R@1,2,5 and V2T R@1,2,5.

Table 21: **Performance gaps, average words and actions/objects complexity of different benchmarks.**

| Benchmark | Avg. Words | Relative Gap (%) | Avg. Objects | Avg. Actions |
|---|---|---|---|---|
| MSVD | 7.0 | -0.404 | 2.34 | 1.56 |
| MSR-VTT | 9.4 | -3.766 | 2.34 | 1.89 |
| DiDeMo | 29.1 | -2.620 | 4.07 | 3.69 |
| VDC short captions | 32.8 | -1.000 | 4.67 | 3.18 |
| DREAM-1K | 59.3 | 1.817 | 5.79 | 6.16 |
| ShareGPT-4o | 125.7 | 0.402 | 8.31 | 5.78 |
| CaReBench | 228.0 | 0.719 | 10.03 | 6.94 |

[†] Objects and actions are extracted by Deepseek V3 (DeepSeek-AI et al., 2024).

```
a boy wearing a red baseball cap
a boy wearing white-brown athletic shoes
a white glove
a black golf club
a vast expanse of grass
a pushcart on the left
some clothing on the pushcart
a deep blue bag on the grass
a railing enclosing the golf course
several yellow-green trees
a white house
a white signboard on the ground

[temporal_objects]
a boy in a red shirt,
a boy in a blue shirt,
a golf club,
a grassy field,
a ball
```

## F  LOGITS VISUALIZATION

To explore how CARE works, we feed its output embedding of a video featuring *a chef is cutting tomatoes in the kitchen* into the last linear layer (i.e. lm_head). It projects the embedding into the vocabulary space. By decode the output logits, we can easily visualize the semantic components of an embedding. It can be discovered that tokens with high logits constitute the essential semantics of the input video, as shown in Figure 5c, describing the main visual objects and actions of the video

Table 22: **Pearson correlation coefficients ($r$) between relative gap and different variables.**

| Item | Pearson $r$ (p-value) |
|---|---|
| Rel Gap vs Avg. Words | +0.706 (p = 0.076) |
| Rel Gap vs Avg. Objects | +0.718 (p = 0.069) |
| Rel Gap vs Avg. Actions | +0.523 (p = 0.229) |

such as *kitchen*, *cutting*, *tomatoes* and *chef*, while the tokens in Figure 5b contain many subwords and irrelevant tokens like *dice*, *car* and *pizza*. It can be inferred that the semantic distribution in the next token space is hugely changed by two-stage SFT, allowing the main semantics to be the core components of the embedding.

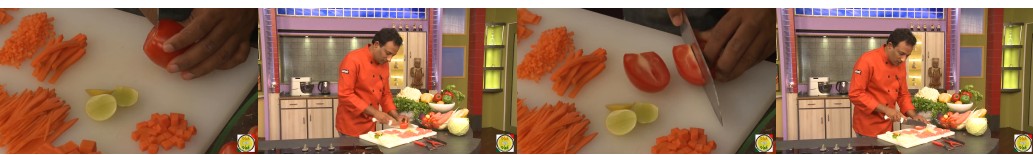

(a) The input video.

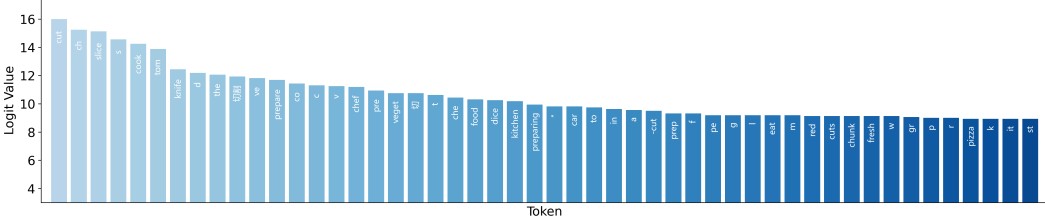

(b) Top 50 tokens decoded from the output embeddings of Qwen2-VL.

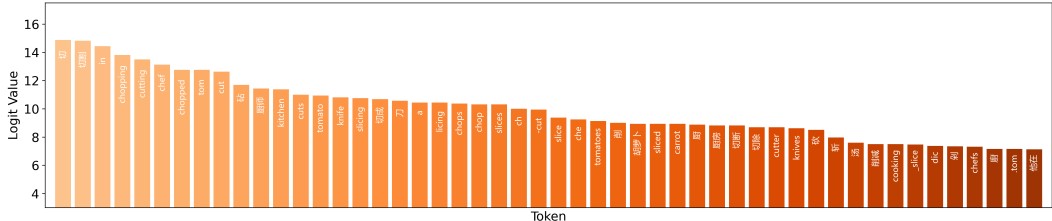

(c) Top 50 tokens decoded from the output embeddings of CARE.

Figure 5: **Top 50 tokens decoded from the output embeddings of Qwen2-VL and CARE.** Qwen2-VL is the baseline model of CARE without any SFT. Compared to Qwen2-VL, two-stage SFT makes the semantic components of CARE embedding much more related to the input video featuring *a chef is cutting tomatoes in the kitchen*.

## G    ANNOTATION GUIDELINES

To inform our annotators the key points that they need to pay attention to, we design a guideline to teach them how to describe videos accurately. The guideline is shown below.

> **Annotation Guideline (Stage 1)**
>
> **Task**
> Your task is to describe videos in detail and hierarchically within 150-300 words. We provide two examples and some points you may need to know.
> **Example 1: Cutting a Watermelon**
> *(A video about cutting a watermelon is provided.)*
>
> - **Summary**  This video shows a man cutting a watermelon.
> - **Object Description**  The man is wearing a green T-shirt and a black apron, with a black mesh hat on his head. His left hand is wearing a gray glove, while his right hand, holding a fruit knife, is wearing a transparent glove. He stands at the corner of the countertop, with a white cutting board in front of him, holding a watermelon. To his left, there is a sink containing another uncut watermelon.
> - **Action Description**  The man first cuts off both ends of the watermelon. Then, he places the watermelon *upright* and *rotate it clockwise*, slicing off the rind piece by

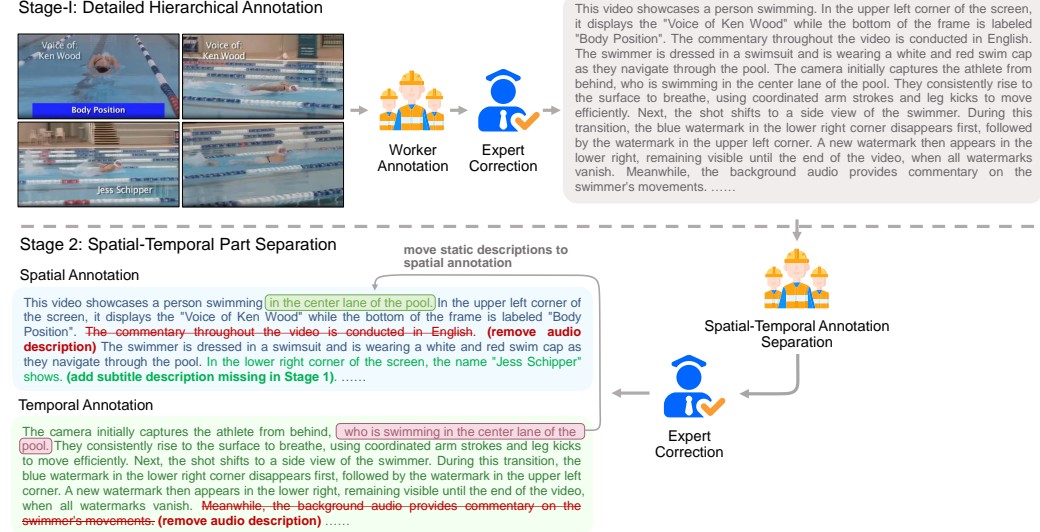

Figure 6: **An overview of the annotation pipeline.** In Stage-I, workers are asked to describe videos hierarchically in detail. In Stage-II, workers need to separate spatial descriptions with temporal descriptions.

piece. He uses the knife to push the rind into a trash bin *on his right*. Next, he takes a light green tray from his right and place it next to the cutting board. After peeling the watermelon, he cuts it into pieces and slides them onto the light green tray.

- **Misc Description** The video is filmed from behind the man, showing a quick and efficient process of cutting the watermelon. With impressive speed, he slices through the fruit, showing his expertise.

**Example 2: Cutting a Tomato**
*(A video about cutting a tomato is provided.)*

- **Summary** In the footage, someone is holding a knife and cutting a tomato on a cutting board.

- **Object Description** The person is wearing black clothes, with a watch on his left wrist. On the cutting board, there are four previously cut tomatoes and one sliced green fruit. On the table, there is a bag of uncut tomatoes and a small knife. *In the top left corner of the video, there is a "luxeat" watermark, and the text "NOW I'VE SEEN EVERYTHING" is written in the bottom left corner.*

- **Action Description** While cutting the tomato, the person first slices it forcefully with one cut, then *speeds up the chopping frequency*, quickly slicing the tomato into neat pieces.

- **Misc Description** The video is filmed from a third-person perspective, showcasing clean and efficient vegetable-cutting. The person's motions are skillful and confident.

**Key Points for Descriptions**

- **Object Description** Describe the entire frame in as much detail as possible. Focus on the objects visible in the frame, clearly describing their positions, appearances, and interactions (e.g "left hand" "right hand" "on the left" "on the right" "above" "below" "upside-down" "holding" "wearing" etc.). This part should follow the description order outlined below: (1) describe the main object in the frames: for example, "The person is wearing a green T-shirt and a black apron, with a black mesh hat on their head. His left hand is wearing a gray glove, while his right hand, holding a fruit knife, is wearing a transparent glove." (2) describe the secondary objects in the frames: for example, "The person is standing at one corner of a metal countertop. In

> front of him is a white cutting board with a watermelon on it. To his left, there is a sink containing another uncut watermelon."
>
> - **Action Description** Clearly describe the actions performed by the main subject, noting the sequence of events (e.g first do X, then do Y). Include details about the nuances of the actions (e.g rotating the watermelon clockwise, flipping it upside-down) and the style of execution (e.g cutting fruit very quickly, climbing a tree clumsily).
>
> - **Misc Description** Describe the video's filming perspective (e.g "first-person," "third-person," "off-site footage of a competition") and provide a brief summary of the overall style and impression conveyed by the actions (e.g orderly and fast watermelon cutting, sharp and efficient movements, clumsy actions, or dangerous behaviors). This part should be concise, within 2-4 sentences.

---

**▌*Annotation Guideline (Stage 2)***

**Task**

In this stage, your task is to separate the original hierarchical descriptions into two parts: spatial descriptions (which do not include any descriptions about movements) and temporal descriptions (which do not include any object descriptions). Camera movements, such as zoom-ins, zoom-outs, etc should be included in temporal descriptions.

**Key Points for Descriptions**

- The spatial description should cover the key objects, secondary objects, and the environment in the frame. It must ensure that, based on the spatial description alone, the videos in the assigned subcategory can be differentiated from one another.

- The temporal description should exclude any obvious static object descriptions that help distinguish different videos. Only the details and sequence of actions should be kept, and it must ensure that, based on the temporal description alone, the videos in the assigned subcategory can be differentiated from one another.

- All the contents of spatial and temporal descriptions should come from the Stage 1 descriptions, and no additional details should be added. Both spatial and temporal descriptions should begin with a summary.

---

## H  LICENSE INFORMATION

**Datasets.** Below are the datasets used in this paper that have known license information: DiDeMo (Hendricks et al., 2017) (BSD 2-Clause License), Activity-Net (Heilbron et al., 2015) (MIT License), DREAM-1K (Wang et al., 2024a) (Apache-2.0 License), VDC (Chai et al., 2024) (Apache-2.0 License). Please note that CAREBENCH will be released with MIT License in the future.

**Models.** Below are the models used in this paper that have known license information: InternVL2 (Chen et al., 2023) (MIT License), InternVL2.5 (Chen et al., 2024) (Qwen License), LLaVA NeXT Video (Zhang et al., 2024b) (Llama 2 Community License), Qwen2-VL (Wang et al., 2024b) (Apache-2.0 License), Tarsier (Wang et al., 2024a) (Apache-2.0 License), LanguageBind (Zhu et al., 2024) (MIT License), Long-CLIP (Zhang et al., 2024a) (Apache-2.0 License), InternVideo2 (Wang et al., 2024d) (Apache-2.0 License).

**Video**

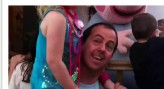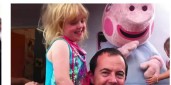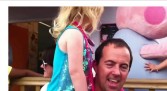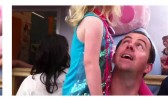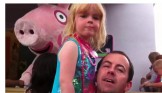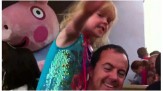

**Caption**

**Annotation:** This video showcases a heartwarming scene at an amusement park where a man is holding a little girl. The man is dressed in a blue top, revealing only his head, neck, and part of his upper body. The little girl has golden hair and is wearing a sleeveless blue top adorned with plenty of sequins on the front. Around her neck, she wears several strands of pink beaded necklaces. Surrounding them are other children and adults, with a person in a Peppa Pig mascot costume standing behind them. The mascot features a pink pig head and a blue body. This costumed character is interacting and waving at the children outside a small fenced area made of wood. Behind them is a white wall that has a blackboard with green and pink patterns drawn on it. The girl is leaning against the man's right arm, being held high by him, with her left hand resting on his neck and her right hand hanging down beside her. She then turns around to look back, releasing her left hand from his neck. The man mouths something to her, and the girl faces the camera again, cheerfully raising her right hand and waving towards it. The Peppa Pig mascot behind them has its left hand resting on its belly and is continuously waving with its right hand, even stopping briefly to embrace someone in front before turning to the right to keep waving. The video captures this scene from the viewpoint of the two characters, and their smiles, along with those of the nearby onlookers, are bright and joyous, showcasing a delightful atmosphere.

**Spatial Annotation:** This video showcases a scene in an amusement park where a man is holding a young girl. The man is dressed in a blue top, revealing only his head, neck, and part of his upper body. The little girl has golden hair and wears a blue sleeveless top adorned with numerous sequins on the front. Around her neck, she sports a necklace made of several pink beads. The girl is leaning against the man's right arm, held high above the ground. Her left hand rests on the man's neck, while her right hand hangs naturally by her side. Surrounding them are other children and adults, and in the background, there's a person dressed in a costume resembling Peppa Pig, with a pink pig head and a blue body. This costumed character is standing in a small enclosed area made of wooden fencing, interacting and greeting the children outside. Behind him is a white wall featuring a small blackboard decorated with green and pink patterns.

**Temporal Annotation:** This video showcases a scene in an amusement park where a man is holding a little girl in his arms. The girl turns her head to look back, releasing her left hand from the man's neck while he says something to her. She then straightens up to face the camera and happily waves her right hand at it. Behind them, a Peppa Pig plush toy stands with its left hand resting on its belly and its right hand waving enthusiastically. At one point, it briefly hugs the person in front before turning to the right to continue waving.

**Video**

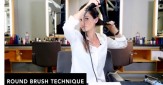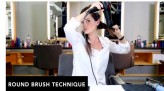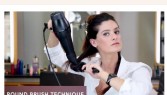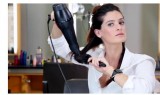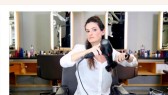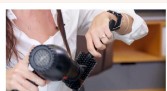

**Caption**

**Annotation:** This video showcases a woman styling her hair. She is dressed in a white blouse and has long hair. On her right wrist, she wears a watch, while her left hand grips a round brush and holds a black hairdryer. In front of her is a white table, which has two black towels draped over it, alongside various combs. The woman is seated on a gray chair, and behind her, there is a row of tables with chairs facing away from her, as well as numerous bottles on the tables. The wall behind her is adorned with several mirrors. At the beginning of the video, she uses the round brush in her left hand to curl a section of her hair on the left side while simultaneously using the hairdryer in her right hand to blow dry those strands. Afterward, she continues to use the round brush to style her hair, securing it at the ends while also using the hairdryer with her left hand to blow dry the hair. The entire video is filmed from a frontal perspective, showcasing her expertise and technique.

**Spatial Annotation:** This video showcases a woman blow-drying her hair. She is dressed in a white top and has long hair. On her right wrist, she wears a watch, while her left hand grips a round hairbrush and holds a black hairdryer. In front of her, there is a white table adorned with two black towels, on which various combs are placed. The woman is seated in a gray chair, with a row of tables and chairs facing away from her behind. The tables are stocked with numerous bottles. Additionally, the wall behind her features several mirrors hanging prominently.

**Temporal Annotation:** This video showcases a woman styling her hair. She starts by using a round brush in her left hand to curl a section of hair on her left side while simultaneously blow-drying it with a hairdryer in her right hand. After that, she continues to use the round brush with her left hand to comb through her hair, securing the brush at the end, and then she uses the hairdryer in her left hand to finish styling those sections of hair.

**Video**

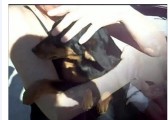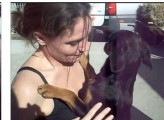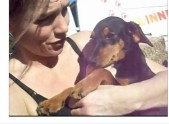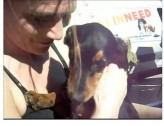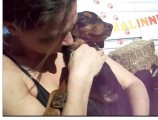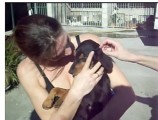

**Caption**

**Annotation:** The video captures the heartwarming moment of a woman embracing her dog. Set outdoors under a brilliant sun, it features a brown-haired woman wearing a black tank top, holding her black dog close. In the background, there's a red and white vehicle adorned with paw print decals. Initially, she gazes down at the side profile of her dog, one arm wrapped around it while the other gently strokes its fur. As the camera rotates clockwise, the dog playfully sticks out its tongue, attempting to lick her. She closes her eyes and turns away, wearing a blissful expression, while both hands continue to caress the dog's neck and head.Later on, she lifts her dog's front paws towards the camera while still scratching its neck. At this moment, another person's arm appears on the right side of the frame, gently rubbing the dog's chin. The woman plants a kiss on the dog's forehead, then leans her head closely against the small pup. The dog tilts its head outward, prompting her to start playing with its front paws using her left hand. She then embraces the dog tightly once more, tenderly stroking the fur on its chin with her right index finger. A man's hand reaches in from the right side of the frame to give the dog some affectionate scratches on its head.As the camera gradually pulls back, the woman continues to stroke the dog's back with her left hand while nuzzling her head against it. The video is shot from a third-person perspective, with the camera positioned very close to the woman and her dog. The scene is filled with the warmth of their embrace, creating a wonderfully intimate atmosphere.

**Spatial Annotation:** The video captures the moment a woman embraces her dog. Set outdoors in glorious sunshine, the scene features a brown-haired woman wearing a black tank top, holding her black dog close. In the background, there is a red and white vehicle adorned with paw print patterns.

**Temporal Annotation:** The video captures the tender moment of a woman embracing her dog. At first, she gazes down at the dog's side profile, with one hand wrapped around the dog and the other gently stroking it. As the camera rotates clockwise, the dog eagerly sticks out its tongue, attempting to lick her, but she closes her eyes and turns away, using both hands to caress the dog's neck and head. Later, she lifts the dog's two front paws to face the camera while continuing to scratch its neck. At this point, another person's arm appears on the right side of the video, reaching out to pet the puppy's chin. The woman kisses the dog's forehead and then presses her head closely against the small dog's. The dog tilts its head outward, and the woman begins to manipulate its front paws with her left hand. She then pulls the dog in tightly, continuing to pet it and gently brushing her right index finger along its chin fur. Just outside the frame on the right, a man extends his hand to pet the dog, scratching its head. As the camera gradually zooms out, the woman uses her left hand to stroke the puppy's back from top to bottom, while also nuzzling her head against its.

**Video**

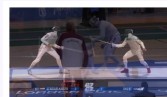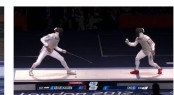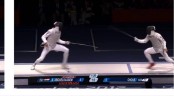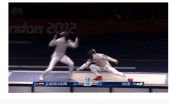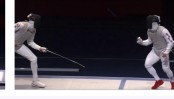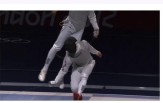

**Caption**

**Annotation:** This video showcases the fencing competition between athletes from the Arab Republic of Egypt and South Korea. At the bottom of the video, you can see the flags of both countries, their respective abbreviations, and the names of the competitors. The match progresses through rounds 1 to 3. On the left side, we have A. ABOUELKASSEM representing the Arab Republic of Egypt, while on the right is South Korean fencer CHOI B. During the match, the Egyptian fencer has their left leg forward and holds the sword in their left hand, while the Korean fencer has their right leg forward and wields the sword in their right hand. Both athletes are clad in fencing uniforms and black helmets, with the South Korean fencer standing out in red shoes. As the match unfolds, they begin by cautiously probing each other before the Korean fencer suddenly lunges forward, striking the Egyptian athlete on the leg. In response, the Egyptian fencer leaps upward to evade the blow but loses their balance upon landing and falls to the left. The second part of the video features a slow-motion replay of this action. The entire video is filmed from the side of the competition area, vividly illustrating the various dynamics of the match.

**Spatial Annotation:** This video showcases the competition between athletes from the Arab Republic of Egypt and South Korea on the fencing arena. At the bottom of the video, you can see the flags of both countries, their abbreviated names, and the names of the athletes. The match is in rounds 1-3. On the left is A. ABOUELKASSEM representing the Arab Republic of Egypt, while on the right is CHOI B. from South Korea. Throughout the competition, both athletes are dressed in fencing attire and wearing black helmets. Notably, the South Korean athlete is wearing red shoes. The Egyptian athlete has their left leg forward and holds the sword in their left hand, while the South Korean athlete has their right leg forward with the sword held in their right hand.

**Temporal Annotation:** This video showcases the competition between the athletes from the Arab Republic of Egypt and South Korea on the fencing arena. During the match, the two players initially engaged in a careful testing of each other's defenses. Suddenly, the South Korean fencer lunged forward with a swift thrust, striking the leg of the athlete from the Arab Republic of Egypt. In response, the Egyptian fencer jumped up, but unfortunately, he lost his balance upon landing and fell to the left. The second part of the video features a slow-motion replay of this sequence of events.

