# OpenReview forum: "CaReBench: A Fine-grained Benchmark for Video Captioning and Retrieval"
_ICLR.cc/2026/Conference — ICLR 2026 Poster_

### Official Review · Reviewer_54Wg · 2025-10-24

**Soundness:** 3
**Presentation:** 2
**Contribution:** 3
**Rating:** 4
**Confidence:** 3

**Summary:**

This paper propose CAREBENCH, a novel evaluation benchmark for fine-grained video captioning and retrieval with manually separated spatial annotations and temporal annotations. New evaluation metrics ReBias and CapST are introduced for a deeper investigation into the spatial and temporal biases inherent in VLMs. Comprehensive experiments are conducted with developed unified baseline method CARE for address both tasks.

**Strengths:**

1.	Fine-grained video captioning and retrieval are worth studying and have certain commonalities. This work proposes a new benchmark for unified evaluation on both tasks.

2.	The experiment result is solid with a newly developed baseline method for both video captioning and retrieval.

3.	Convincing visualization examples are provided to prove the effectiveness of the proposed method.

**Weaknesses:**

1.	Although video captioning and retrieval tasks have a certain duality, the name of the proposed fine-grained video captioning and retrieval benchmark in this article seems to simply blend these two tasks, appearing to be an improvement and integration of previous work, lacking some innovative new concepts of video understanding.

2.	The developed baseline method is encouraged to be tested on some other related video understanding benchmarks to demonstrate its generalization ability.

**Questions:**

1.	See weakness.

2.	I appreciate the dedicated efforts on the proposed benchmark and its experiment results. However, I still have some doubts about the positioning of this job as a benchmark or dataset. To begin with, as a benchmark, its targeted video captioning and retrieval are relatively old video understanding tasks, the name of the proposed method should require more innovative new concepts of video understanding. Meanwhile, many compared benchmarks in Table 1 are the test set of a larger video understanding dataset, which includes more training samples for the cross-validation of its effectiveness, but this work only proposes a testing benchmark instead of a complete dataset for fine-grained video captioning and retrieval. In summary, I would give a much higher score if this work proposed a complete dataset with both a training set and a testing benchmark. If not, I would consider raising my score if the author integrates these two tasks to provide convincing and innovative concepts of video understanding.

---

> ### Author Response · Authors · 2025-11-28
> **Response to Reviewer 54Wg**
>
> We sincerely thank you for your time and constructive review, which has prompted us to critically re-examine the positioning and significance of our work. We address your concerns below:
>
> > W1. The proposed fine-grained video captioning and retrieval benchmark in this article seems to simply blend these two tasks
>
> We respectfully clarify that the core contribution of CaReBench is not merely blending two "old tasks", but pioneering a fine-grained evaluation paradigm for MLLMs. The novelty resides in two layers:
>
> **1. Fine-grained Human-annotated Caption**
>
> Traditional benchmarks (MSR-VTT, MSVD, etc.) feature short captions (avg. <20 words, see Figure 4) that fail to probe modern MLLMs’ capacity limits. CaReBench is the first benchmark to introduce spatiotemporally decoupled, fine-grained human-annotated captions for captioning & retrieval. This design is not a simple task fusion but a diagnostic instrument: via ReBias, we quantitatively expose for a temporal bias in all major VLMs (temporal bias exceeding spatial bias by 15–30% in existing models).
>
> **2. Methodological Insight**
>
> *(a) Both tasks operate on the same underlying output space of MLLMs.*
>
> Under the next-token prediction (NTP) paradigm, retrieval and captioning differ only in how the hidden representations are accessed. Retrieval uses an EOL prompt to compress the semantics of a video or text into the hidden state of the next first token, which functions as a compact embedding. Captioning, in contrast, uses a "describe the video in detail" prompt, leveraging the hidden representations of subsequent tokens to decode a full description. Despite this difference in usage, both tasks fundamentally rely on aligning the model’s hidden space with the semantics of the original input—making a unified formulation both natural and theoretically consistent.
>
> *(b) Empirical evidence shows that our two-stage training explicitly strengthens and reshapes this shared output space.*
>
> As shown in Appendix F, we visualize and semantically decode the logits of CaRe’s next-token output. Before training, Qwen2-VL’s decoded next-token semantics correlate weakly with the actual video content; after our two-stage training, CaRe’s next-token distributions concentrate heavily on vocabulary directly relevant to each video. This confirms that our method explicitly restructures the output space to better reflect video semantics.
>
> Furthermore, the two tasks quantitatively reinforce each other. As reported in Section 5.4 and Table 6, adding retrieval adaptation to the baseline (R@1: 25.6→77.0) improves captioning performance (+1.4 F1, 26.8→28.2). Adding fine-grained caption adaptation on top (F1: 28.2→33.4) further boosts retrieval performance (+1.0 R@1, 77.0→78.0). This mutual gain—together with the logits visualization—supports our core motivation: retrieval and captioning benefit from being treated as two manifestations of the same underlying semantic alignment problem.
>
> > W2. Lack of experiments about CaRe's generalization ability
>
> Thanks for your suggestion. We provide two out-of-domain tests to show CaRe's generalization ability.
>
> **(1) General video understanding benchmarks**
>
> To validate CaRe's general video understanding ability, we tested CaRe and other popular models on TVBench and MVBench. Please see global responses for the results. The results show that CaRe demonstrates strong generalization on general video understanding benchmarks (60.4 on MVBench and 50.1 on TVBench).
>
> **(2) Fine-grained benchmarks with varying caption length and action-object complexity**
>
> We evaluate CaRe vs Qwen2-VL on 7 benchmarks with different caption length and action-object complexity (Appendix D). The correlation coefficients of their performance gap vs avg. words (+0.706, p = 0.076), vs avg. objects (+0.718, p = 0.069) and vs avg. actions (+0.523, p = 0.229) indicate that CaRe demonstrates great performance on long-text, fine-grained tasks—particularly those involving numerous actions and objects. For shorter captions with simpler content, the model remains effective, though the improvement is naturally more modest given its specialization for fine-grained scenarios.
>
> > Q1. Doubts about benchmark/dataset positioning
>
> **(1) Positioning: CaReBench as a "Challenging Test Set"**
>
> Our hierarchical captions are intentionally detailed, covering objects, actions, OCR text, special effects, shot transitions, etc.—precisely the granularity needed to probe the capability frontiers of VLMs. The annotation costs 10-20 minutes/video with about 50 annotators involved in, making large-scale human-annotated training sets prohibitively expensive. Therefore, we prioritized building a high-quality challenge test set over quantity.
>
> **(2) Building a training set**
>
> Considering the cost of building a large training dataset,  we are now focusing on using CaRe model to semi-automatically expand a large-scale dataset. We will release our training dataset to the community as soon as it is ready.

---

### Official Review · Reviewer_DL3n · 2025-10-29

**Soundness:** 3
**Presentation:** 3
**Contribution:** 3
**Rating:** 6
**Confidence:** 4

**Summary:**

The paper introduces CaReBench, a fine-grained benchmark for video captioning and retrieval with human-annotated spatial and temporal descriptions, along with two new evaluation metrics, ReBias and CapST, to analyze spatiotemporal bias. It further proposes CARe, a unified two-stage model that jointly handles both tasks, achieving competitive performance compared to CLIP-based and MLLM-based baselines.

**Strengths:**

1. Proposes CaReBench, a fine-grained benchmark with detailed spatial and temporal annotations and new metrics ReBias and CapST for analyzing spatiotemporal bias.
2. Introduces a unified model CARe that jointly handles video captioning and retrieval through a two-stage training framework.
3. Demonstrates strong experimental results and clear analysis, supported by high-quality human annotations and well-presented methodology.

**Weaknesses:**

See questions.

**Questions:**

1. Could the authors clarify the motivation for integrating video captioning and video retrieval into a single unified model? It would be helpful to explain whether these two tasks bring any mutual benefits or complementary effects in practice.
2. Regarding the use of LLMs for NLI judgment, could the authors elaborate on its reliability? For instance, would the evaluation results vary significantly if a different LLM were used?
3. It would be interesting to know whether the EOL-based embedding extraction method can effectively distinguish long texts that are semantically similar but not identical.

---

> ### Author Response · Authors · 2025-11-28
> **Response to Reviewer DL3n**
>
> Thank you for your valuable time and constructive reviews. We would like to address your concerns below. We would appreciate it if you let us know whether our revisions adequately address your concerns.
>
> > Q1. Motivation for integrating video captioning and video retrieval into a single unified model
>
> Our motivation for unifying retrieval and captioning tasks and using MLLMs for retrieval tasks are listed below:
>
> **(1) Both tasks operate on the same underlying output space of MLLMs.**
>
> Under the next-token prediction (NTP) paradigm, retrieval and captioning differ only in how the hidden representations are accessed. Retrieval uses an EOL prompt to compress the semantics of a video or text into the hidden state of the next first token, which functions as a compact embedding. Captioning, in contrast, uses a “describe the video in detail” prompt, leveraging the hidden representations of subsequent tokens to decode a full description. Despite this difference in usage, both tasks fundamentally rely on aligning the model’s hidden space with the semantics of the original input—making a unified formulation both natural and theoretically consistent.
>
> **(2) Empirical evidence shows that our two-stage training explicitly strengthens and reshapes this shared output space.**
>
> As shown in Appendix F, we visualize and semantically decode the logits of CaRe’s next-token output. Before training, Qwen2-VL’s decoded next-token semantics correlate weakly with the actual video content; after our two-stage training, CaRe’s next-token distributions concentrate heavily on vocabulary directly relevant to each video. This confirms that our method explicitly restructures the output space to better reflect video semantics.
>
> Furthermore, the two tasks quantitatively reinforce each other. As reported in Section 5.4 and Table 6, adding retrieval adaptation to the baseline (R@1: 25.6→77.0) improves captioning performance (+1.4 F1, 26.8→28.2). Adding fine-grained caption adaptation on top (F1: 28.2→33.4) further boosts retrieval performance (+1.0 R@1, 77.0→78.0). This mutual gain—together with the logits visualization—supports our core motivation: retrieval and captioning benefit from being treated as two manifestations of the same underlying semantic alignment problem.
>
> > Q2. Reliability of NLI judgment.
>
> We evaluated all the models in the paper with three LLM judges—DeepSeek-V3, Gemini-2.5-Pro and GPT-4.1—and report the details in the global response. Pair-wise Pearson correlations between the resulting model scores all exceed 0.96, quantitatively verifying that CapST remains stable and reliable regardless of the chosen judge.
>
> > Q3. It would be interesting to know whether the EOL-based embedding extraction method can effectively distinguish long texts that are semantically similar but not identical.
>
> The embeddings of long texts are discriminative after EOL-based training. We here provide an example, each comprising five long sentences (A, B, C, D, E) with similar structure and length. Sentences A and B are semantically highly similar, while A's semantic similarity to C, D, and E progressively decreases. The embedding similarities between sentences, computed via the EOL prompt embedding method, are also shown below.
>
> *(A)  Base Sentence*
>
> The video shows a woman jogging along a riverside path, keeping a steady and rhythmic pace as she moves past a long row of benches, briefly adjusting her smartwatch without slowing down, and slightly accelerating when approaching the bridge where several cyclists pass by.
>
> *(B) Highly Similar (cosine similarity with A: 0.8945)*
>
> The video shows a woman running along a riverside path, maintaining a consistent cadence while passing multiple benches, glancing at her watch without breaking her pace, and subtly increasing her speed as she gets closer to the bridge where other people are crossing.
>
> *(C) Subtle Contradiction, Still Structurally Similar (cosine similarity with A: 0.7578)*
>
> The video shows a woman moving along the riverside path at a progressively slower pace, hesitating near the benches as if deciding whether to rest, checking her watch with a concerned expression, and almost coming to a full stop upon reaching the bridge, showing no intention to accelerate.
>
> *(D) Medium Difference from Sentence A (cosine similarity with A: 0.4863)*
>
> The video shows a woman walking her dog beside the river, pausing near each bench to let the dog sniff around, pulling out her phone to read a message, and waiting at the foot of the bridge until the dog is ready to continue strolling.
>
> *(E) Large Difference from Sentence A (cosine similarity with A: 0.4980)*
>
> The video shows a woman sitting on the grass near the riverside, reading a paperback novel while occasionally lifting her head to observe passing joggers, stretching her legs while staying seated, and remaining in the same spot throughout the clip without any forward movement.

---

### Official Review · Reviewer_uxnZ · 2025-10-29

**Soundness:** 4
**Presentation:** 3
**Contribution:** 4
**Rating:** 10
**Confidence:** 3

**Summary:**

The paper introduces CAREBENCH, a new benchmark of 1,000 videos with rich human-written captions designed to evaluate fine-grained video understanding. Each video has hierarchical, multi-aspect annotations (overall summary, static object details, dynamic action details, and miscellaneous context). Spatial and temporal descriptions are manually separated to probe how models handle static scene vs. dynamic action information. Based on this data, the authors propose two evaluation metrics: ReBias, which measures a model’s bias toward spatial vs. temporal retrieval, and CapST, an LLM-based captioning score combining recall/precision of objects and events. They also present CARE, a unified video-language model (built on Qwen2-VL) trained via a two-stage fine-tuning (first to improve caption detail, then contrastively for retrieval). Experiments (zero-shot evaluation on CAREBENCH and other datasets) show CARE achieves very strong performance in both detailed captioning and video retrieval, outperforming specialized retrieval models (CLIP variants) and powerful captioning MLLMs. The paper carefully discusses related work, conducts ablations of the two-stage training, and analyzes a consistent spatiotemporal bias (models do much better on spatial cues than temporal actions).

**Strengths:**

1. CAREBENCH demonstrates notable strengths across key areas. Its originality lies in the novel hierarchical annotation schema—dividing video content into summary, objects, actions, and miscellaneous categories—which addresses a critical gap in video understanding evaluation. The introduction of specialized ReBias and CapST metrics further showcases innovation, leveraging the dataset's structure to examine biases and caption quality.

2. The work exhibits strong methodological rigor through its careful human annotation process involving dual annotators and expert refinement. The two-stage training approach is well-designed, with Stage I focusing on fine-grained caption alignment and Stage II on contrastive retrieval training, a design validated through ablation studies.

3. The paper is clearly written and logically structured, with informative figures and tables that effectively support the technical content. The comprehensive related work discussion demonstrates deep scholarly engagement.

4. In terms of significance, CAREBENCH provides valuable tools that reveal how current VLMs often rely on spatial shortcuts while struggling with dynamic actions. By exposing these biases, the work should stimulate development of more balanced models, with the CARE baseline demonstrating the feasibility of unified modeling approaches.

**Weaknesses:**

1. The work has several limitations that warrant consideration. While the paper effectively identifies significant spatiotemporal biases in models, it stops at measurement and does not propose methods to mitigate these biases. This focus on benchmarking over developing corrective techniques somewhat limits its immediate practical impact on improving model design.

2. The generalization capability of the proposed CARE model remains uncertain. Its evaluation on standard benchmarks beyond CAREBENCH shows mixed results, particularly on MSR-VTT retrieval where it fails to clearly outperform established baselines. More comprehensive analysis is needed to understand how well the unified approach transfers across diverse domains.

3. The CapST metric relies exclusively on DeepSeek-V3 as the sole LLM judge for evaluating objects and actions, which raises potential concerns about robustness and inherent biases in the assessment. Although some validation against human judgment is provided, a more extensive evaluation using multiple LLMs, varied prompts, or additional human raters would strengthen confidence in the metric's reliability and general applicability.

**Questions:**

1. How sensitive is the CapST metric to the choice of LLM judge or prompt? Have you validated its consistency using alternative models like GPT-4 or through prompt variations?

2. While CARE shows strong zero-shot retrieval results on MSR-VTT, MSVD, and DiDeMo, could you provide more experimental details? How does it perform on captioning tasks for these datasets, and what are its limitations on out-of-domain videos?

3. Given CAREBENCH's scale of 1,000 videos, do you recommend it primarily for evaluation or also for fine-tuning? What are the risks of overfitting and how might training on it affect model robustness?

4. Will the benchmark be publicly released? Are there licensing restrictions from FineAction, and how will reproducibility be ensured?

5. Given the spatial bias identified by ReBias, have you explored any mitigation strategies, such as data augmentation or balanced training? What approaches would you suggest to reduce this bias?

6. How does CapST handle paraphrases or partial omissions? Could you provide quantitative correlation between CapST scores and human judgments to further validate the metric?

7. Could you elaborate on how conflicts between annotators were resolved during expert refinement? What guidelines ensured consistency, and was any inter-annotator agreement measured?

---

> ### Author Response · Authors · 2025-11-28
> **Response to Reviewer uxnZ (Part 1)**
>
> We sincerely appreciate the time and effort you have invested in reviewing our submission. Please allow us to address your concern below:
>
> > Q1. CapST metric stability
>
> We reported CapST on GPT-4.1, DeepSeek-V3, and Gemini-2.5-Pro and evaluated pairwise Pearson correlations. With All inter-judge correlations larger than 0.96, the ablation study demonstrats that CapST remains highly stable across different LLM judges. For mode details, please see our global responses about LLM judge ablation.
>
> > Q2.1 CaRe retrieval experimental details
>
> All our models are evaluated on MSR-VTT, MSVD, and DiDeMo under identical settings to ensure fairness. The detailed settings are as follows:
>
> *(a) Data preparation:* For MSR-VTT, we adopt the standard 1K-A split protocol, which was introduced in JSFusion and has since become the de facto benchmark split in the text-video retrieval field. For MSVD, we used the official caption annotations for evaluation. For DiDeMo, we adopted the paragraph retrieval method commonly used in works such as InternVideo2, where multiple captions of the same video are concatenated as the ground-truth caption.
>
> *(b) Inference:* All models used 32-frame video inputs. After embedding all videos and texts in the dataset, we computed cosine similarities between video and text embedding matrix and obtained recall@1, 5, 10 based on similarity rankings.
>
> > Q2.2 Captioning tasks for these datasets
>
> Thanks for your valuable request! We need to kindly clarify that it is difficult to effectively evaluate fine-grained understanding models on MSR-VTT, MSVD, and DiDeMo due to their limitations:
>
> (1) Their video captions are overly brief and cannot comprehensively cover all details in the videos, making them unsuitable for evaluating the fine-grained video caption performance of strong MLLMs.
>
> (2) They rely on n-gram-based metrics (e.g., CIDEr, BLEU) for caption evaluation. N-gram metrics primarily rely on token-level matching and have lost their effectiveness for evaluating long–long or short–long text pairs, as investigated in Appendix C.
>
> > Q2.3 Limitations on out-of-domain videos and mixed results on benchmarks like MSR-VTT.
>
> For out-of-domain videos, we conducted tests on the following two domains:
>
> **(1) General video understanding benchmarks**
>
> To test CaRe's general video understanding capability, we evaluated CaRe on MVBench and TVBench. Please see global responses for the results. The results show that CaRe demonstrates strong generalization on general video understanding benchmarks (60.4 on MVBench and 50.1 on TVBench).
>
> **(2) Fine-grained benchmarks with varying caption length and action-object complexity**
>
> We evaluate CaRe vs Qwen2-VL on 7 benchmarks like ShareGPT-4o, MSVD and MSR-VTT with different caption length and action-object complexity (Appendix D). The correlation coefficients of their performance gap vs avg. words (+0.706, p = 0.076), avg. objects (+0.718, p = 0.069) and avg. actions (+0.523, p = 0.229) indicate that CaRe demonstrates great performance on fine-grained tasks (CaReBench, ShareGPT-4o, etc.)—particularly those involving numerous actions and objects. For shorter captions with simpler content (MSVD, MSR-VTT, etc.), the model remains effective, though the improvement is naturally more modest given its specialization for fine-grained scenarios.
>
> > Q3. Do you recommend CaReBench primarily for evaluation or also for fine-tuning?
>
> Thank you for this insightful suggestion. We recommend that the benchmark be used primarily for **evaluation purposes**. The captions are intentionally detailed, covering objects, actions, OCR text, special effects, shot transitions, etc.—precisely the granularity needed to probe the capability frontiers of VLMs.  For training, we suggest using CaRe model to semi-automatically expand a large-scale dataset, which is the focus of our future work.
>
> > Q4. Will the benchmark be publicly released? Are there licensing restrictions from FineAction, and how will reproducibility be ensured?
>
> Yes. We will publicly release it with MIT license since there is no license restriction from FineAction. We will open-source the CaReBench evaluation toolkit—identical to the scripts used in our paper—together with the JSON inference outputs of all the models to reproduce all reported results. The toolkit provides extensible interfaces so that new models can be plugged in with minimal effort.
>
> > Q5. Given the spatial bias identified by ReBias, have you explored any mitigation strategies, such as data augmentation or balanced training? What approaches would you suggest to reduce this bias?
>
> This work provides two insights to the community: the possibility of unifying retrieval and captioning, and the spatiotemporal understanding imbalance in VLMs. Our baseline primarily addresses the first aspect, while the second is actively being investigated in our ongoing work. We hope these insights inspire the community to develop more spatiotemporally balanced training recipes.

---

> ### Author Response · Authors · 2025-11-28
> **Response to Reviewer uxnZ (part 2)**
>
> > Q6.1 How does CapST handle paraphrases or partial omissions?
>
> CapST relies on LLM judges to handle (1) paraphrases and cross-sentence references, and (2) partial omissions of object attributes in model predictions. All examples come from the evaluation of CaRe 7B.
>
> **(1) LLM-Based Inference for Paraphrases and References**
>
> *Example A: Pronoun Resolution*
> ```json
> {
>  "gt": "...the child is seated in a barber chair...",
>  "pred": "...a woman is shaving the head of a boy...",
>  "events_pred": [
>    {
>      "event": "A boy sits in a chair.",
>      "relationship": "entailment",
>      "reason": "The video description states that 'the child is seated in a barber chair' and refers to the child as 'he', which implies the child is a boy."
>    }
>  ]
> }
> ```
> The pronoun "he" and the explicit word "boy" are bridged, and the relation is marked as entailment even though the original GT string never literally uses "boy".
>
> *Example B: Synonyms for Motion*
> ```json
> {
>  "events_gt": [
>    {
>      "event": "Athlete draws back javelin",
>      "relationship": "entailment",
>      "reason": "The description states 'the man then swings his arm back and prepares to release the javelin', which is synonymous with drawing the javelin back."
>    }
>  ]
> }
> ```
> Even without lexical identity ("draw back" vs. "swings his arm back"), the LLM connects the motion semantics.
>
> (2) Single-Attribute Object Extraction for Partial Omissions
>
> When the prediction omits part of a complex object description, a naïve one-line comparison would fail entirely. As mentioned in Section 3.4.2,  we instruct the LLM to split attributes during extraction. This approach preserves partial credit: if the prediction captures only some attributes, those components still receive entailment.
>
> *Example C: Outdoor Athlete*
>
> ```json
> {
>  "gt": "...male athlete is dressed in a blue sports jacket... black athletic shorts... white sneakers... standing on a red running track...",
>  "objects_gt": [
>    {
>      "object": "There is a male athlete in a blue sports jacket.",
>      "relationship": "entailment",
>      "reason": "The description explicitly states there is 'a man in a blue jacket'..."
>    },
>    {
>      "object": "There is a male athlete in black athletic shorts.",
>      "relationship": "entailment",
>      "reason": "The description explicitly states the man is wearing 'black shorts'..."
>    },
>    {
>      "object": "There is a male athlete in white sneakers.",
>      "relationship": "neutral",
>      "reason": "The video description does not mention the man's footwear or the color of his shoes."
>    },
>    {
>      "object": "There is a red track.",
>      "relationship": "entailment",
>      "reason": "The description explicitly states the scene is 'on a red track field'."
>    }
>  ]
> }
> ```
>
> Here the prediction confirms both the jacket and shorts but says nothing about sneakers. With single-attribute splitting, the scoring achieves two hits out of three clothing attributes (about 66.7%) rather than zero; if these elements are merged into "a blue jacket, black shorts, and white sneakers," the missing footwear detail will make the entire object neutral.
>
> > Q6.2 Could you provide quantitative correlation between CapST scores and human judgments to further validate the metric?
>
> We invited human experts to perform "which-is-better" evaluations on our benchmark. Pearson correlation coefficients between these metrics and Elo scores (i.e., human preferences) indicate that Action F1 and Object F1 exhibit strong correlations (r = 0.81 and r = 0.71, respectively), suggesting they effectively capture human preferences. For more details, please see Appendix C.
>
> > Q7.1 Could you elaborate on how conflicts between annotators were resolved during expert refinement?
>
> Upon completion of the first-stage annotation, experts review the two annotations for each video and then merge them. During the merging phase, two types of conflicts may emerge: factual and linguistic.
>
> (1) *Factual conflicts* typically arise when rare annotation errors overlooked during expert review, resulting in contradictory factual statements between the two annotations of the same video. In the merging stage, experts resolve these conflicts by verifying them against the video content and retaining only the correct parts.
>
> (2) *Linguistic conflicts* typically stem from differences in linguistic expression among annotators. Experts reconcile these variations to produce a final annotation that is fluent and comprehensive.
>
> > Q7.2 What guidelines ensured consistency, and was any inter-annotator agreement measured
>
> Before starting annotation, we provided a detailed annotation guideline (refer to Appendix G) to our annotators. This guideline comprehensively outlined what content should be included in each part of the hierarchical annotation. We began annotation only after reaching consensus on the guideline among the annotators. This ensured quality of annotations.

---

### Official Review · Reviewer_X3xc · 2025-10-30

**Soundness:** 3
**Presentation:** 3
**Contribution:** 3
**Rating:** 6
**Confidence:** 4

**Summary:**

This paper introduces CaReBench, a new fine-grained benchmark for video captioning and retrieval, comprising 1,000 videos with human-annotated hierarchical captions. Each caption includes spatial and temporal components, allowing a detailed evaluation of models’ understanding of static objects and dynamic actions.

The authors also propose two novel metrics:
ReBias, for quantifying spatiotemporal bias in retrieval tasks, and
CapST, for evaluating captions with both precision and recall over spatial and temporal elements.

Additionally, the paper presents CARE, a unified baseline model built upon Qwen2-VL, trained via two-stage SFT to handle both captioning and retrieval tasks simultaneously.

**Strengths:**

1. **Innovative Benchmark Design:**
CAREBENCH introduces a uniquely structured dataset with hierarchical annotations that explicitly separate spatial and temporal descriptions, filling a clear gap in current benchmarks.

2. **New Evaluation Metrics:**
ReBias and CapST are well-motivated and address the shortcomings of existing metrics (e.g., CIDEr, AutoDQ, VDCScore), providing more interpretable and fine-grained evaluations.

3. **Unified Framework:**
The CARE model elegantly unifies retrieval and captioning within one architecture using a two-stage fine-tuning pipeline, demonstrating potential task synergy.

4. **Comprehensive Experiments:**
The paper includes extensive quantitative results across state-of-the-art models and thorough ablation studies on the two-stage fine-tuning process.

**Weaknesses:**

1. **Reliance on LLM-Based Evaluation:**
CapST depends on an LLM (DeepSeek-V3) as the evaluator, which may introduce bias. The paper lacks inter-rater consistency checks or human alignment experiments.

2. **Theoretical Depth:**
The paper mainly focuses on empirical contributions. The “unified mapping” idea (ϕ: RT×H×W×C → RD) is intriguing but not theoretically explored.

3. **Incomplete Bias Mitigation:**
While ReBias reveals spatiotemporal imbalance, the proposed model still shows clear temporal bias, and the paper does not attempt to mitigate it.

4. **Ablation Detail and Visualization:**
Some analyses, such as the effects of video length, motion complexity, or per-category performance variance, are missing and would add valuable insight.

**Questions:**

1. How stable are CapST scores across different LLM evaluators (e.g., GPT-4o vs DeepSeek-V3)?

2. Have you explored joint multi-task training (caption + retrieval simultaneously) instead of the sequential two-stage pipeline?

3. What are the main annotation challenges in separating spatial and temporal components, and how is annotator consistency ensured?

4. Given the rapid evolution of multimodal large language models (MLLMs), have you considered adding comparisons with more recent MLLMs (for example, Qwen2.5-VL) to demonstrate how your benchmark and unified model stack up against the latest models?

---

> ### Author Response · Authors · 2025-11-28
> **Response to Reviewer X3xc (Part 1)**
>
> Thank you for your insightful comments and valuable suggestions, which have significantly contributed to improving the manuscript. Below, we provide detailed responses to each of the questions you raised. Please let us know whether our revisions adequately address your concerns.
>
> > Q1. How stable are CapST scores across different LLM evaluators (e.g., GPT-4o vs DeepSeek-V3)?
>
> We computed CapST on GPT-4.1, DeepSeek-V3, and Gemini-2.5-Pro and evaluated pairwise Pearson correlations. All inter-judge correlations exceed 0.96, indicating that CapST remains highly stable across different judges. For more details, please see the global responses.
>
> > Q2. Have you explored joint multi-task training (caption + retrieval simultaneously) instead of the sequential two-stage pipeline?
>
> We appreciate this insightful suggestion, which we also considered during the development of this work. The second-stage contrastive learning needs a very large batch size (e.g., 768 or larger) to ensure sufficient intra-batch negative diversity, a prerequisite for strong retrieval performance. Integrating the first-stage captioning objective into this pipeline under such batch sizes, however, far exceeds our current resource constraints. We are actively investigating specialized training strategies and optimization techniques to enable stable joint training, and we will report any positive findings in a future update.
>
> > Q3.1 the main annotation challenges in separating captions, and how is annotator consistency ensured?
>
> The main difficulty lies in (i) separating spatial objects from temporal actions without leaking redundant information, and (ii) ensuring each caption remains discriminative even among highly similar videos. We address this through two key designs:
>
> **(1) Curating visually similar videos to enforce discriminability.**
>
> As described in Sec. 3.1, we intentionally select 10–20 highly similar videos within each FineAction sub-category. This guarantees that intra-class videos are highly similar yet still distinguishable, whereas inter-class diversity remains large, naturally increasing the challenge level of CaReBench.
>
> **(2) Expert verification to ensure minimality and discriminability.**
>
> After Stage-II separation, experts review all captions within the same sub-category to (i) remove redundant objects/actions, and (ii) check that each caption uniquely corresponds to its video. This expert pass guarantees that only the necessary spatial/temporal cues are preserved while maintaining strong intra-class discriminability.
>
> > Q3.2 How is annotator consistency ensured?
>
> To ensure annotator consistency, we rely on a structured guideline and a consistent expert-driven refinement process:
>
> **(1) A detailed, operational annotation guideline.**
>
> Annotators follow a clear definition of spatial vs. temporal content, along with concrete examples (Appendix G). This guideline standardizes the treatment of object attributes, action sequences, and the exclusion of irrelevant information, ensuring consistency across annotators.
>
> **(2) Expert auditing that strictly applies the same guideline.**
>
> Two independent annotations are merged for each video and refined by an expert who applies the same rules as the guideline. Misplaced details (e.g., static backgrounds within temporal captions) are relocated or removed.

---

> ### Author Response · Authors · 2025-11-28
> **Response to Reviewer X3xc (Part 2)**
>
> > Q4. Given the rapid evolution of multimodal large language models (MLLMs), have you considered adding comparisons with more recent MLLMs (for example, Qwen2.5-VL) to demonstrate how your benchmark and unified model stack up against the latest models?
>
> Thank you for this helpful suggestion. We evaluated Qwen2.5-VL and demonstrated below:
>
> *Table 1. CaReBench General Retrieval*
> |**Model**|**T2V R@1**|**R@5**|**R@10**|**V2T R@1**|**R@5**|**R@10**|
> |-|-|-|-|-|-|-|
> |Qwen2-VL 7B|76.6|95.3|**98.7**|*77.4*|95.6|*98.7*|
> |Qwen2.5-VL 7B|**77.5**|*95.4*|*98.6*|75.5|*95.8*|*98.7*|
> |**CaRe**|*77.0*|**95.6**|**98.7**|**79.0**|**96.8**|**99.1**|
>
> *Table 2. CaReBench Spatial/Temporal Retrieval*
> |**Model**|**S-T2V R@1**|**R@5**|**R@10**|**S-V2T R@1**|**R@5**|**R@10**|**T-T2V R@1**|**R@5**|**R@10**|**T-V2T R@1**|**R@5**|**R@10**|
> |-|-|-|-|-|-|-|-|-|-|-|-|-|
> |Qwen2-VL 7B|**78.2**|*95.5*|98.5|*75.4*|*95.0*|*98.1*|**51.9**|*84.8*|**94.9**|*52.7*|*85.4*|**95.2**|
> |Qwen2.5-VL 7B|*77.0*|*95.6*|**98.8**|72.6|94.8|97.8|51.4|84.1|93.9|50.5|83.1|93.5|
> |**CaRe**|76.8|**96.3**|*98.7*|**78.1**|**95.8**|**99.3**|*50.7*|**85.3**|*94.4*|**53.4**|**86.3**|*94.0*|
>
> *Table 3. CaReBench Caption (Events)*
> |**Model**|**# Params**|**Personal Care**|**Social & Relax**|**Sports & Exercise**|**Household**|**Overall**|
> |-|-|-|-|-|-|-|
> |Qwen2-VL|7B|28.4/23.9/34.9|27.5/20.8/40.3|33.0/26.6/43.6|25.7/20.2/35.1|28.8/22.9/39.0|
> |Qwen2.5-VL|7B|*30.0*/21.2/*51.0*|29.7/*21.3*/ *48.9*|36.1/28.0/*50.8*|27.2/19.4/**45.6**|31.1/22.7/49.2|
> |**CaRe (stage-I)**|7B|*33.9*/ *25.4*/50.8|**32.4**/**24.0**/**49.8**|**42.8**/**33.7**/**58.5**|**31.5**/**24.4**/44.7|**35.3**/**26.9**/*51.3*|
> |**CaRe**|7B|**34.4**/**25.6**/**52.6**|*32.2*/**24.0**/48.8|*42.3*/*33.3*/ *58.1*|*30.9*/ *23.4*/ *45.3*|*35.1*/ *26.6*/**51.4**|
>
> *Table 4. CaReBench Caption (Objects)*
> |**Model**|**# Params**|**Personal Care**|**Social & Relax**|**Sports & Exercise**|**Household**|**Overall**|
> |-|-|-|-|-|-|-|
> |Qwen2-VL|7B|23.7/15.8/47.7|23.0/15.1/47.8|24.9/16.2/53.1|24.8/16.8/47.2|24.0/15.9/49.1|
> |Qwen2.5-VL|7B|**32.8**/**25.3**/46.6|**32.7**/**25.9**/44.2|**34.8**/**27.6**/47.3|**34.0**/**27.7**/44.1|**33.5**/**26.5**/45.5|
> |**CaRe (stage-I)**|7B|*32.1*/ *22.6*/ *55.3*|*31.3*/ *22.2*/ *53.1*|*33.2*/ *23.2*/ *58.4*|*33.6*/ *23.8*/ **57.1**|*32.4*/ *22.9*/ *55.7*|
> |**CaRe**|7B|30.9/21.1/**57.2**|*31.5*/21.9/**55.6**|31.8/21.3/**62.6**|32.6/23.0/*55.8*|31.7/21.8/**57.8**|
>
> As we can see, Qwen2.5-VL represents a clear advance over its predecessor (e.g., Qwen2-VL), and the updated results show that CaRe-7B still retains a consistent margin on both video caption and retrieval.

---

### Author Response · Authors · 2025-11-28
**Global Response**

Dear ACs and reviewers,

Thank you very much for your invaluable time and insightful comments on our submission. Here we provide a global response.

First, we would like to kindly highlight our contributions.

- **Benchmark** CaReBench is the first fine-grained video-understanding benchmark with spatiotemoral decoupled captions. The 1,000 clips are manually annotated by human annotators, yielding detailed hierarchical captions. We propose two new metrics CapST and ReBias to comprehensively reveal VLMs’ performance and biases by separately evaluating spatial and temporal tasks.
- **Baseline** We propose CaRe, a single MLLM that unifies retrieval and captioning. CaRe shows competitive results in fine-grained video captioning and retrieval, and has great generalization across different fine-grained benchmarks.

For the most common questions, here are our responses:

**Summary**

**1. CapST remains highly stable & reliable across different LLM judges (inter-judge correlations>0.96). (Reviewer X3xc, uxnZ and DL3n)**

**2. CapST demonstrate significant correlations with the human Elo score (Action F1 0.81, Object F1 0.71), suggesting they capture human preferences very well. (Reviewer X3xc, uxnZ and  DL3n)**

**3. CaRe shows strong generalization on general video understanding benchmarks such as MVBench and TVBench. (Reviewer uxnZ and 54Wg)**

> Q1. CapST scores stability across different LLM evaluators

To show the stability of CapST across different LLM judges, we evaluated all the models using an identical prompt  on Deepseek V3 (our papar implementation), Gemini 2.5 Pro and GPT 4.1.

For a intuitive comparison, we computed the correlations between these LLM judges. All inter-judge correlations exceed 0.96, indicating that **CapST remains highly stable across different judges.** The results are shown below.

*Table 1. CapST (GPT 4.1)*
|Models|# Params|Action F1|Action R|Action P|Object F1|Object R|Object P|
|-|-|-|-|-|-|-|-|
|GPT-4o mini|-|36.1|30.5|44.2|34.0|26.7|46.6|
|LLaVA NeXT Video|7B|25.9|20.2|36.2|24.5|18.1|37.6|
|InternVL2|7B|23.9|21.4|27.2|23.5|18.3|32.7|
|InternVL 2.5|7B|27.3|21.0|38.8|30.1|***24.9***|37.9|
|InternVL 2.5|72B|30.0|22.9|43.6|31.6|**26.3**|39.5|
|MiniCPM-V 2.6|7B|32.7|25.0|47.1|30.6|22.4|48.0|
|Tarsier|7B|27.2|19.0|48.1|32.0|24.2|47.4|
|Qwen2-VL|7B|28.5|24.6|33.9|22.9|15.8|41.7|
|Qwen2-VL|72B|29.9|23.0|42.7|23.2|15.6|45.5|
|CaRe Stage 1|7B|**35.3**|**26.9**|**51.0**|**32.5**|23.0|***53.2***|
|CaRe|7B|***34.4***|***26.1***|***50.3***|***32.3***|23.0|**54.2**|

*Table 2. CapST (Gemini 2.5 Pro)*
|Models|# Params|Action F1|Action R|Action P|Object F1|Object R|Object P|
|-|-|-|-|-|-|-|-|
|GPT-4o mini|-|41.5|31.7|59.9|39.9|30.7|57.0|
|LLaVA NeXT Video|7B|29.0|19.9|53.2|30.1|21.5|50.1|
|InternVL2|7B|28.6|23.1|37.4|28.5|21.4|42.8|
|InternVL 2.5|7B|30.4|21.1|53.9|35.6|28.7|47.1|
|InternVL 2.5|72B|33.4|23.4|58.4|36.8|**29.5**|48.9|
|MiniCPM-V 2.6|7B|35.3|24.7|62.2|36.4|26.1|60.2|
|Tarsier|7B|32.1|21.5|**63.6**|36.4|27.1|55.4|
|Qwen2-VL|7B|32.8|25.1|44.0|27.7|18.4|56.2|
|Qwen2-VL|72B|34.8|26.1|52.1|28.1|18.4|60.0|
|CaRe Stage 1|7B|**41.2**|**31.5**|59.6|**37.1**|26.0|***64.8***|
|CaRe|7B|***40.3***|***30.2***|60.7|36.4|25.1|**66.6**|

*Table 3. Pearson Corr. DeepSeek V3 vs GPT-4.1*
|Metric|Correlation|P-value|
|-|-|-|
|Action F1|0.97|5.6e-07|
|Action R|0.96|3.6e-06|
|Action P|0.94|1.8e-05|
|Object F1|0.99|1.3e-08|
|Object R|0.99|2.4e-09|
|Object P|0.96|5.6e-06|
|Avg|0.97|-|

*Table 4. Pearson Corr. DeepSeek V3 vs Gemini-2.5-Pro*
|Metric|Correlation|P-value|
|-|-|-|
|Action F1|0.98|1.5e-07|
|Action R|0.97|5.4e-07|
|Action P|0.95|7.9e-06|
|Object F1|0.98|1.5e-07|
|Object R|0.99|2.4e-08|
|Object P|0.99|3.1e-10|
|Avg|0.98|-|

*Table 5. Pearson Corr. GPT-4.1 vs Gemini-2.5-Pro*
|Metric|Correlation|P-value|
|-|-|-|
|Action F1|0.98|3.5e-07|
|Action R|0.92|7.7e-05|
|Action P|0.92|4.9e-05|
|Object F1|0.99|3.5e-09|
|Object R|0.99|1.2e-09|
|Object P|0.96|2.7e-06 |
|Avg|0.96|-|

> Q2. human alignment experiments of CapST

We conduct human-aligned validation to show that CapST reflects fine-grained caption quality and correlates with human judgment (Appendix C). Specifically, we invite human experts to perform ”which-is-better” evaluations on our benchmark. Pearson correlation coefficients between these metrics and Elo scores (i.e. human preference). The results indicate that: **CapST demonstrate significant correlations with the Elo score (Action F1 0.81, Object F1 0.71), suggesting they capture human preferences very well.**

> Q3. generalization of CaRe

To demonstrate CaRe's generalization capability on out-of-domain tasks, we conduct video QA testing on MVBench and TVBench. Data for models other than CaRe are from the official TVBench report. The results show that **CaRe demonstrates strong generalization on general video understanding benchmarks.**

|Models|MVBench|TVBench|
|-|-|-|
|GPT-4o|47.8|35.8|
|VideoChat2|51.0|33.0|
|PLLaVA-7B| 46.6| 34.2|
|Gemini 1.5 Pro|**60.5**|*46.5*|
|CaRe-7B|*60.4*|**50.1**|

---

### Meta-Review · Area_Chair_tGYV · 2026-01-07

**Summary:**

The reviews reflect a generally positive consensus that the paper offers a valuable benchmark, novel metrics, and a carefully executed unified model, with strong empirical support and clear exposition.

**Reviewer Concerns:**

The reviewers expressed concerns about evaluation robustness, theoretical depth, and the focus on measurement rather than mitigation, the authors’ detailed responses and additional experiments substantially address these points and reduce the associated risks. Taken together, the reviewers view the contribution as solid and useful to the community despite some remaining limitations.

**Reviewer Scores:**

Finally, I recommend to accept this paper.

---

### Decision · Program_Chairs · 2026-01-26

Accept (Poster)